# Evolution of volatility and composition in sesquiterpene-mixed and α-pinene secondary organic aerosol particles during isothermal evaporation

Zijun Li[1], Angela Buchholz[1, *], Arttu Ylisirniö[1], Luis Barreira[1,2], Liqing Hao[1], Siegfried Schobesberger[1], Taina Yli-Juuti[1], Annele Virtanen[1, *]

[1]Department of Applied Physics, University of Eastern Finland, Kuopio, Finland

[2]Atmospheric Composition Research, Finnish Meteorological Institute, Helsinki, Finland

*Correspondence to*: Angela Buchholz (angela.buchholz@uef.fi), and Annele Virtanen (annele.virtanen@uef.fi)

**Abstract.** Efforts have been spent on investigating the isothermal evaporation of α-pinene SOA particles at ranges of conditions and decoupling the impacts of viscosity and volatility on evaporation. However, little is known about the evaporation behavior of SOA particles from biogenic organic compounds other than α-pinene. In this study, we investigated the isothermal evaporation behavior of the α-pinene and sesquiterpene mixture (SQTmix) SOA particles under a series of relative humidity (RH) conditions. With a set of in-situ instruments, we monitored the evolution of particle size, volatility, and composition during evaporation. Our finding demonstrates that the SQTmix SOA particles evaporated slower than the α-pinene ones at any set of RH (expressed with the volume fraction remaining (VFR)), which is primarily due to their lower volatility and possibly aided by higher viscosity under dry conditions. We further applied positive matrix factorization (PMF) to the thermal desorption data containing volatility and composition information. Analyzing the net change ratios (NCRs) of each PMF-resolved factor, we can quantitatively compare how each sample factor evolves with increasing evaporation time/RH. When sufficient particulate water content was present in either SOA system, the most volatile sample factor was primarily lost via evaporation, and changes in the other sample factors were mainly governed by aqueous-phase processes. The evolution of each sample factor of the SQTmix SOA particles was controlled by a single type of process, whereas for the α-pinene SOA particles it was regulated by multiple processes. As indicated by the coevolution of VFR and NCR, the effect of aqueous-phase processes could vary from one to another according to particle type, sample factors, and evaporation timescale.

## 1 Introduction

Atmospheric oxidation of volatile organic compounds (VOCs) can lead to a complex mixture of condensable organic vapors spanning ranges of functionalities and structures, and hence volatilities (Hallquist et al., 2009). Parts of these organics contribute to the mass concentration of secondary organic aerosol (SOA) particles. Gas-particle partitioning is a dynamic process of importance, influencing the composition in the gas and particle phase as well as the atmospheric lifetime of SOA. For a long time, gas-particle partitioning has been considered as a near-instantaneous process (Odum et al., 1996; Donahue et al., 2006), under the assumptions that SOA particles consist mainly of intermediate/semi-volatile compounds and exists in a liquid state. Recent measurements suggest that SOA particles consist of large amounts of organic compounds with low or extremely low volatilities

(Cappa and Jimenez, 2010; Ehn et al., 2014; Mohr et al., 2019) and that particles can adopt viscous semisolid or amorphous solid states (Virtanen et al., 2010; Pajunoja et al., 2013; Zhang et al., 2015). All this emerging evidence challenges the abovementioned assumptions, which underlie the treatment of SOA with the partitioning theory. When volatilities of organic compounds range from intermediate to extremely low volatile (Donahue et al., 2012), the equilibration timescales of phase partitioning span from seconds to hours in liquid particles (Shiraiwa and Seinfeld, 2012). In viscous particles, bulk diffusion limitations can increase these equilibration timescales to the order of years (Li and Shiraiwa, 2019).

Monoterpenes ($C_{10}H_{16}$) are the most abundant terpene emissions in boreal forests (Tarvainen et al., 2007; Bäck et al., 2012), driving SOA formation and growth in the atmosphere (O'Dowd et al., 2002; Jokinen et al., 2015). As the most representative monoterpene, α-pinene has been widely used to generate SOA as a proxy for boreal forest SOA. SOA yield studies using environmental chambers have suggested that α-pinene SOA particles are dominated by semi-volatile organic compounds (Pathak et al., 2007; Shilling et al., 2008). But multiple studies which investigated the isothermal evaporation of α-pinene SOA particles at a range of relative humidity (RH) consistently demonstrated that SOA particles do not evaporate as rapidly as expected for semi-volatile organic mixtures (Vaden et al., 2011; Wilson et al., 2014; Yli-Juuti et al., 2017; D'Ambro et al., 2018). These findings suggest the importance of unaccounted low-volatility organic compounds, particle phase reactions, and viscous phase states (Vaden et al., 2011; Wilson et al., 2014; Yli-Juuti et al., 2017; D'Ambro et al., 2018). While volatility distributions of organic compounds mainly determine the extent to which particles evaporate at high RH, diffusion limitations attributed to particle viscosity significantly hinder particle evaporation under dry conditions. Recent studies have also explored the oxidation and temperature dependence of the evaporation of α-pinene derived SOA particles. For instance, increasing the oxygen-to-carbon ratio (O:C) of the initial particles reduces the particle evaporation rate and possibly induces aqueous-phase processes which form low volatility compounds especially for highly oxidized SOA particles (Buchholz et al., 2019). Decreasing temperature can suppress particle evaporation by lowering the saturation vapor concentrations (C*) of the SOA compounds and/or increasing particle bulk viscosity (Shiraiwa et al., 2017; Li et al., 2019).

Efforts have been spent on investigating the evaporation of α-pinene SOA particles, but the diversity of VOC emissions from trees and the complexity of particulate constituents complicate the description of organic vapor partitioning in boreal forests. Branch enclosure measurements with boreal tree species have revealed that VOC emission profiles vary in terpene species and ratios, dependent on seasons (Hakola et al., 2017) or degrees of abiotic/biotic stress (Zhao et al., 2017a; Kari et al., 2019). Laboratory studies have shown that compared to α-pinene derived SOA particles, those derived from oxidizing sesquiterpenes ($C_{15}H_{24}$) or actual (stressed) Scots pine emissions feature distinct properties, in terms of mass yield, volatility, and molecular composition (Faiola et al., 2018; Ylisirniö et al., 2020). Given these observations, it is necessary to investigate the evaporation behavior of SOA particles derived from terpene precursors other than α-pinene and even from real plant emissions. Current measurements have identified that large amounts of farnesenes and bisabolenes are emitted from boreal tree species (Hakola et al., 2017; Danielsson et al., 2019) and that their derived SOA are of potential climate significance by influencing cloud formation (Mentel et al., 2013; Zhao et al., 2017a).

To facilitate a better understanding of biogenic organic vapor partitioning in boreal forests, α-pinene and a sesquiterpene mixture were chosen as precursors to generate two different types of biogenic SOA particles for isothermal evaporation under a range of RH conditions at room temperature. The mixture consists of farnesenes and bisabolenes, which are acyclic and monocyclic sesquiterpenes, respectively. The aim of this study is to compare the evaporation behavior of sesquiterpene derived SOA particles

to that of α–pinene derived SOA particles. For this, both the particle size changes as well as the particle composition evolution were measured, and their differences and similarities will be discussed.

## 2 Methods

### 2.1 Experimental setup

Two different types of biogenic SOA particles were generated in a 13 L oxidation flow reactor (OFR) (Kang et al., 2007; Lambe et al., 2011) for isothermal evaporation experiments taking place at a wide range of RH at 25 °C. The experimental setup and procedure were similar to our previous evaporation studies (Yli-Juuti et al., 2017; Buchholz et al., 2019; Li et al., 2019) and a detailed description of our experimental setup can be found in the Supplement. Briefly, the experimental sequence consisted of biogenic SOA production, followed by particle size selection with simultaneous dilution of the gas phase, and humidity-controlled isothermal particle evaporation.

Either α-pinene (Sigma-Aldrich, 98%) or a sesquiterpene mixture (Sigma-Aldrich, mixture of isomers) was introduced into a heated $N_2$ flow with a syringe pump system (Kari et al., 2018). Farnesene isomers (40%, acyclic) and bisabolene isomers (40%, monocyclic) are the two dominant species in the mixture of sesquiterpenes, followed by other unidentified sesquiterpenes (Ylisirniö et al., 2020). The VOC-containing flow was then mixed with a humidified flow of $N_2$ and $O_3$. Overall, 5 L min$^{-1}$ of total flow containing VOCs (254 − 261 ppb) and $O_3$ (13.01 − 13.40 ppm) with RH of 41% − 44% was introduced into the OFR for photooxidation at controlled temperature (~25 °C). Under the illumination of 254-nm UV lamps, hydroxyl radicals (OH) were produced from the reaction of water vapor with O ($^1$D) which was generated from photolysis of $O_3$. We produced α-pinene and sesquiterpene mixture (SQTmix) SOA with comparable oxidation conditions. The OH exposure ranges from 0.9 to 2.6×10$^{11}$ molec cm$^{-3}$ as calculated with the OFR model (Peng et al., 2015; Peng et al., 2016) which takes the external OH reactivity into account. The elemental composition of SOA particles was characterized by a high-resolution time-of-flight aerosol mass spectrometer (HR-ToF-AMS, Aerodyne Research Inc.). It should be noted that rather than by pure photooxidation, SOA was formed via both ozonolysis and photooxidation reactions, as $O_3$ levels of over 1 ppm were used. For all evaporation experiments of one SOA system, the aerosol mass concertation in the OFR was very similar. Assuming a particle density of 1.4 g cm$^{-3}$, the mass concentrations of polydisperse α-pinene and SQTmix SOA from the OFR were estimated to be 399 ± 16 and 128 ± 16 µg m$^{-3}$, respectively. It has been found that compounds with C* of 0.1 µg m$^{-3}$ and below dominate the SOA composition in a previous study using the same type of SOA (Ylisirniö et al., 2020). Even though the aerosol mass concentration in the OFR in our study is higher than the typical ambient level by one order of magnitude, such difference would not affect the gas-particle partitioning behavior of compounds with C* $\leq 0.1$ µg m$^{-3}$. Experimental conditions and results for the SOA generation are summarized in Table S1.

The generated SOA was introduced into two parallel nanometer aerosol differential mobility analyzers (NanoDMA, model 3085, TSI) for particle size selection. The size selection process also diluted the organic vapors by two orders of magnitude with an open-loop sheath flow and thereby initiated particle evaporation. To vary the RH in the samples, we humidified/dried the sheath flow of the NanoDMAs. The desired RH was set to one of three conditions: dry (< 7% RH), intermediate (40% RH), or high (80% RH). Eventually, a narrow distribution of SOA particles with 80 nm electrical mobility diameter was fed (i) to bypass lines with varying lengths for short evaporation measurements of up to 3 min, or (ii) to a 25 L stainless-steel residence time chambers (RTC) for intermediate evaporation measurements of up to 40 min with 10 min intervals, or (iii) to a 100 L RTC for long evaporation

measurements of up to 7.5 h with 1 h intervals. Prior to each particle evaporation experiment, the NanoDMAs, bypass tubing and
105 RTCs were flushed for at least 12 h with purified air at the desired RH of the following experiment.

**2.2 Characterization of particle evaporation**

Size changes of SOA particles due to evaporation were periodically monitored using a scanning mobility particle sizer (SMPS, model 3080, TSI). The extent of particle evaporation was evaluated in the terms of volume fraction remaining (VFR). Assuming particles are spherical, the VFR was calculated as follows:

$$VFR = (\frac{D_{p,t}}{D_{p,0}})^3 \ , \tag{1}$$

where $D_{p,0}$ and $D_{p,t}$ are the particle sizes measured at the start (i.e., as selected by the NanoDMAs) and after time $t$ of evaporation, respectively. The temporal evolution of particle evaporation was illustrated by plotting VFR against residence time ($t_R$) in the bypass tubing or RTC, defined as "evapogram", as shown in Figure 1. The selected particle size was calibrated using dry ammonium sulfate particles.

The thermal desorption behavior and chemical composition of particle samples were characterized using a chemical ionization mass spectrometer (CIMS, Aerodyne Research Inc.) coupled with a custom-built Filter Inlet for Gases and AEROsols (FIGAERO) (Ylisirniö et al., 2021) using iodide-adduct ionization (Lopez-Hilfiker et al., 2014). The operation of FIGAERO-CIMS can be found in the Supplement. Particle samples were collected for analysis (i) right after size selection (fresh, avg. $t_R$ = 0.25 h, due to the 0.5-h collection times) and (ii) after isothermal evaporation in the RTC (RTC, avg. $t_R$ = 4.25 h). After a 30-min sample collection,
the collected particles were gradually desorbed with a heated $N_2$ flow of which the temperature was firstly ramped from 25 °C to ~200 °C within 20 min (desorption period), and then maintained at above 190 °C for an additional 15 min (soak period) to evaporate any residual organics left on the filter. The relationship between the temperature of the maximum desorption signal ($T_{max}$) of a single compound and its $C^*$ was calibrated against a set of polyethylene glycol compounds (PEG, PEG 4 – 8) (Ylisirniö et al., 2021) with known vapor pressures (Krieger et al., 2018). The desorption temperature ($T_{desorp}$) range is divided into three volatility
ranges (i.e. semi-volatile organic compounds (SVOC), low-volatility organic compounds (LVOC), and extremely low volatility organic compounds (ELVOC)) as defined by Donahue et al. (2012).

The desorption temperature-dependent change in the sum of the organic signals over the temperature range is referred to as sum thermogram, STG. The appearance of the STG depends on the number of molecules collected on the FIGAERO filter and the volatility distribution of the sample. We are interested in determining if some compounds in the particle phase are lost or produced
during isothermal evaporation. To be able to investigate this, we need to account for changes in the STG due to different collected sample mass and the isothermal evaporation. As it was not possible to determine the collected sample mass independently, we normalize the $STG(T)$ with the total ion signal of each sample ($N_{Tot}$):

$$STG_N(T) = \frac{STG(T)}{N_{Tot}} \ . \tag{2}$$

In addition, we need to take into account how much material is expected to be removed from each individual particle due to the
135 isothermal evaporation. We assume that this removal is proportional to the change in the average VFR ($VFR_{avg}$) determined for the corresponding evaporation time and can be described with the removal factor ($f_{removal}$):

$$f_{removal} = \frac{VFR_{avg,RTC}}{VFR_{avg,fresh}} \cdot \alpha_{MW_{avg}}^{-1} \cdot \beta_{\rho_{avg}} \ , \tag{3}$$

where $VFR_{avg,fresh}$ and $VFR_{avg,RTC}$ are the average VFR during the FIGAERO sampling time at fresh and RTC evaporation stages. $\alpha_{MW_{avg}}$ is a parameter that describes the relative change in the signal-weighted average molecular weight (MW) of the particle bulk, $MW_{avg,RTC}/MW_{avg,fresh}$, and $\beta_{\rho_{avg}}$ is a parameter that captures the relative change in the average particle density, $\rho_{avg}$, $\rho_{avg,RTC}/\rho_{avg,fresh}$. These two parameters convert the isothermal evaporation effect from the volumetric base to the molecular base.

We scale the normalized STG for the RTC sample ($STG_{N,RTC}(T)$) with $f_{removal}$ expressed in Eq. (3) to obtain the scaled STG for the RTC sample ($STG_{SC,RTC}(T)$):

$$STG_{SC,RTC}(T) = STG_{N,RTC}(T) \cdot \frac{VFR_{avg,RTC}}{VFR_{avg,fresh}} \cdot \alpha_{MW_{avg}}^{-1} \cdot \beta_{\rho_{avg}} \qquad (4)$$

A more detailed justification for this approach can be found in Appendix A. The values of $\alpha_{MW_{avg}}$ and $\beta_{\rho_{avg}}$ which were used for the calculation of $STG_{SC,RTC}(T)$ are given in Table C1. The ratio of $VFR_{avg}$ is proportional to the material loss per particle, so is the resulting $STG_{SC,RTC}(T)$. Hence $STG_{SC,RTC}(T)$ and $STG_{N,fresh}(T)$ can be compared quantitatively (Figure 2a), and the differences between them directly indicate if compounds with a certain desorption temperature are lost, produced or remained unchanged during the isothermal evaporation. A similar approach can be used to investigate the evolution of PMF factors as explained in section 3.3.

Previously, T$_{max}$ of the *STG(T)* was used to compare the overall volatility between particle samples (Ylisirniö et al., 2021). Here, the median desorption temperature (T$_{50}$, at which half of the cumulative *STG(T)* signal is reached) was used instead because it is a more general measure of the overall desorption behavior. Typically, these T$_{50}$ values were higher than the T$_{max}$ values, as most signals were recorded at temperatures above T$_{max}$.

## 2.3 Deconvolution of FIGAERO-CIMS data set with Positive Matrix Factorization (PMF)

Since it was introduced by Paatero and Tapper (1994), PMF has been widely used to identify the contribution of different sources of trace compounds in ambient measurements (Ulbrich et al., 2009; Zhang et al., 2011; Yan et al., 2016). More recently, PMF has been adapted to analyze laboratory experiments for understanding chemical or physical aspects of systems of interest (Craven et al., 2012; Zhao et al., 2017b; Buchholz et al., 2020). Regarding a FIGAERO-CIMS data set, PMF can separate sample signals from filter background and contamination. But more than that, this method can also identify multiple factors which represent not only isomeric compounds with different volatilities but also thermally decomposed products for each ion. Following the procedure outlined in Buchholz et al. (2020), constant error values (CNerror) which were derived from the noise at the end of thermogram scans were applied to all ions without further down-weighing. The PMF results were calculated using the PMF Evaluation Tool (PET 3.05) with one to twelve factors and five fpeak rotations from -1 to +1. Additional information about the PMF analysis is described in the Supplement including the justification for the selected solution. The PMF analysis was applied independently for each precursor to sets of FIGAERO-CIMS samples. Each set represents particles from one SOA precursor (α-pinene or SQTmix), which were collected at both evaporation stages (fresh and RTC) under dry and high RH conditions. Two types of blank measurements were added to the data set: (i) Measurements of the clean FIGAERO filter without sampling from the setup. These blanks characterize the overall instrument background. (ii) Measurements of filters sampled directly after size selection for 30 min but with the NanoDMA voltage set to 0 V. These blanks represent the background due to, e.g., adsorption of remaining gas-phase compounds onto the filter during the normal sample collection procedure.

## 3 Results and Discussion

### 3.1 Bulk volatility of SOA particles

#### 3.1.1 Isothermal evaporation behaviour of SOA particles

The isothermal evaporation behavior of the SOA particles is illustrated by the VFR as a function of $t_R$ in Figure 1. The evaporation rate of the dry SOA particles was the slowest, and the differences in the sum (Figure 2) and factor thermograms (Figure 3 and Figure 4) between two evaporation stages were minor. The particle evaporation rate became faster with increasing RH for both SOA systems. When particulate water was present, the contribution of compounds in the SVOC range was reduced during fresh stages (Figure 2). As shown in previous studies (Yli-Juuti et al., 2017; Buchholz et al., 2019; Li et al., 2019; Zaveri et al., 2020), considerable kinetic limitations exist for the evaporation of volatile compounds in this type of dry SOA particles due to the substantially high viscosity. Particulate water reduces the viscosity and thus enhances particle evaporation with increasing RH. The comparable evaporation rates under intermediate and high RH conditions suggest that particle evaporation can be approximated as a liquid-like process for both conditions (i.e., at RH ≥ 40%), But in addition to this plasticizing effect, particulate water content may also induce aqueous-phase processes during isothermal evaporation (Buchholz et al., 2019; Petters et al., 2020). For the investigated SOA particles, we observed strong evidence of such processes under high RH conditions (RH = 80%). These are detailed in section 3.3.3. Quantifying the effects of particle viscosity and aqueous-phase processes on the SOA particle evaporation would require developing detailed processes models considering particle phase chemistry, which is not the primary focus of this study.

At any set RH, the evaporation rate of the SQTmix SOA particles was slower than that of the α-pinene ones, although both SOA were produced under comparable oxidation conditions. Such distinguishable evaporation patterns are most likely driven by (i) the distinct particulate volatility distributions jointly controlled by molecular weight and functionality, expressed by elemental composition as a proxy (Li et al., 2016), and/or (ii) the possible differences in particle bulk viscosities especially under dry conditions.

#### 3.1.2 Thermal desorption behaviour of SOA particles

In Figure 2, the thermal desorption behavior of particle samples which were collected at fresh (avg. $t_R$ = 0.25 h) and RTC (avg. $t_R$ = 4.25 h) evaporation stages under dry (RH < 7%) and high RH (RH 80%) conditions are displayed as normalized $STG(T)$ ($STG_{N,fresh}(T)$, solid line) and scaled ones ($STG_{SC,RTC}(T)$, dashed line), respectively (Figure 2a, b). These two types of $STG(T)$ together are hereinafter referred to as STGs for simplicity unless otherwise specified. The corresponding $T_{50}$ and $VFR_{avg}$ are shown in Figure 2c, and the sampling periods for FIGAERO-CIMS thermograms are highlighted with colored areas in Figure 1. For each SOA system of interest, similar mass concentration of organic material after size selection was ensured for both dry and high RH conditions so that the volatility distribution of compounds in the condensed phase was not significantly affected. For the α-pinene case, the mass concentration of organic material after size selection under dry and high RH conditions were 4.47 and 5.31 μg cm$^{-3}$, respectively. For the SQTmix case, the corresponding values were 0.97 and 1.39 μg cm$^{-3}$ under dry and high RH conditions.

Compared to the STGs of the fresh samples, the STGs of the RTC samples shifted to higher $T_{desorp}$ values with increases in $T_{50}$, regardless of the RH conditions. When examining the particle desorption profiles (i.e., the STGs), we note that the removal of compounds which were thermally desorbed below 120 °C, and the corresponding changes in $T_{50}$, were more pronounced between

the fresh and RTC samples at high RH as compared to those under dry condition. Such difference in the changes of STGs between two RH conditions agrees with our observation of faster particle evaporation rates in the presence of water (see Figure 1).

Under dry conditions, a larger fraction of LVOC and ELVOC (collectively (E)LVOC) contributed to the STGs of the SQTmix SOA particles, with higher values of $T_{50}$ when compared to the α-pinene particles (Figure 2a, c). Consistent with the changes in $VFR_{avg}$ under dry conditions, relatively less increase in $T_{50}$ and decrease in the STGs were observed in the SQTmix SOA particles as well. On the other hand, similar STGs were observed for the fresh samples at high RH, regardless of SOA particle type. According to the evaporation model simulations described in a previous study using a similar measurement setup (Li et al., 2019),

a majority of I/SVOC is expected to evaporate rapidly from fresh particles during the first 8 – 30 min at high RH. It should be noted that during the same evaporation timescale (≤ 0.5h), the evaporation of (E)LVOC is expected to be negligible. Therefore, the $VFR_{avg}$ ($t_R$ = 0.25 h) is approximately determined by the ratio of (IVOC + SVOC)/(LVOC + ELVOC) in the initial particles. As the FIGAERO sampling periods last for 30 min, it follows that under high RH conditions, the fresh particles lost a significant fraction of the initially present I/SVOC during sample collection. Thus, the similarity in STGs between the α-pinene and SQTmix

SOA particles suggests that the (E)LVOC fraction in both SOA types had a similar volatility distribution and/or thermal desorption behavior. Note that this does not mean that the same types of compounds were present in the two different SOA types. For the same reason, the difference in $T_{50}$ between the two different types of fresh particles was less noticeable than the difference in $VFR_{avg}$ at high RH (Figure 2c, solid circles).

### 3.2 PMF factors of SOA particles

Depending on the RH conditions or SOA precursors, the particle size and volatility appeared to evolve differently during isothermal evaporation (Figure 1 and Figure 2). To better assess the compositional and volatility changes of the investigated SOA particles, we performed PMF analyses to deconvolute the thermal desorption data. Each derived factor constitutes a group of organic compounds with very similar "temporal" behavior. The PMF algorithm does not prescribe any meaning to the position of a value in the dataset, i.e., the $T_{desorp}$ or desorption time values are only used to define the order of the data points. When volatility acts as

the primary factor driving the composition change in the particles, compounds with similar desorption behavior correlate and are grouped into factors. In each factor, compounds of similar volatility evaporate in a similar manner during the isothermal evaporation so that the shape of the factor thermogram remains more or less constant between conditions. However, the occurrence of aqueous-phase processes may complicate the grouping of compounds especially for highly oxidized samples (Buchholz et al., 2020). Compounds with somewhat different volatility may no longer be separated but rather be grouped together due to how they

are affected by the aqueous phase. This can create changes in the appearance of the factor thermogram (e.g., broadening) and possibly induce a non-negligible shift in $T_{desorp}$ (≥ 15 °C) dependent on the extent of aqueous-phase processes. We provide more details about the behavior of the PMF algorithm, how compounds are grouped, and why the shape and characteristic $T_{desorp}$ may change in the Supplement (see Section S1.2.3).

Two types of factors were identified. Factors occurring in particle samples but predominantly in filter blank measurements are

240 defined as type B ("background") factors. The sum of type B factors showed similar absolute signal strength regardless of sample types. But while this contributed 10 – 60% to the total sum signal of the particle samples, it accounted for more than 80% of the total sum signal in filter blank samples. Type B factors displayed either nearly constant or very shallow factor thermograms. Factors which showed contributions in particle samples but not in filter blank samples were assumed to describe the collected particle sample and thus defined as type F ("sample") factors. In Buchholz et al. (2020), these sample factors were distinguished into ones

dominated by direct desorption of compounds (type V) and those dominated by products of thermal decomposition (type D). The careful analysis of the sample factors in this study showed that we could not make such a strict distinction. Thus, we decided to use the terms background factor (type B) and sample factor (type F) and point out which of the sample factors showed strong signs of thermal decomposition products.

PMF solutions with eight and ten factors were chosen for the α-pinene and SQTmix SOA particles, respectively. In both PMF results, five factors are assigned as sample factors and the rest are considered background factors (i.e., type B factors). In the following discussion (Figure 3 and Figure 4), type B factors and the blank measurements are omitted. All mass spectral profiles and all factor thermograms of all samples of each data set can be found in Figure S3 and S4. Furthermore, ion distributions and bulk properties are visualized for each sample factor in the form of modified Kroll diagrams (Kroll et al., 2011) in Figure S10 and S11 by plotting the average carbon oxidation state (OSc) versus the carbon number (Cnum). By lumping ions with the same carbon number into a grid with a 0.2-interval on the y-axis of OSc, the issue of overlapping signals was avoided.

### 3.2.1 α-pinene SOA particles

In total, five sample factors (AF1 – 5, colored) were identified for the α-pinene SOA particles as shown in Figure 3. For AF1 – AF4, average MW increased with higher $T_{50}$ (i.e., lower volatility). While these factors were dominated by compounds with $C \leq$ 10, as expected for a precursor composition of $C_{10}H_{16}$, additional amounts of compounds with $C > 10$ (i.e., dimers/oligomers) contributed to the total signal of AF3 and especially to that of AF4 (see also Figure S10a). With increasing $T_{50}$ values, factors had longer carbon chain lengths and higher oxygen contents, as indicated by their average molecular composition. There was no clear association between OSc and $T_{50}$ for factors AF1 - AF4, since the increase in carbon chain lengths was counterbalanced by the simultaneous addition of oxygen and hydrogen numbers. Therefore, the decrease in volatility of type V factors was mainly driven by the increase in average MW.

For AF5, its bulk properties and composition distribution (Figure 3 and Figure 5a) were closest to those of AF2 and AF3 with compounds with MW < 200 Da dominating their factor mass spectra. However, the thermal desorption behavior of AF5 was completely different with almost all of its signal occurring at $T_{desorp} > 100$ °C and a continuous increase with $T_{desorp}$ until the soak period started. Many of the compounds assigned to AF5 also showed contributions to other factors at lower $T_{desorp}$ values. It is very unlikely that all these were isomeric compounds spanning 5 or more orders of magnitude in C* between the isomeric forms. It is much more probable that those compounds with small MW in AF5 were decomposition products of thermally unstable compounds with larger MW and lower volatility (D'Ambro et al., 2018; Schobesberger et al., 2018; Yang et al., 2021).

### 3.2.2 SQTmix SOA particles

In a similar way as for the α-pinene SOA particles, five sample factors (i.e., SF1 – 5, colored) were identified for the SQTmix SOA particles, as shown in Figure 4. For SF1 – SF4, lower volatilities characterized by higher $T_{50}$ values again correlated with increasing average MW but not with average OSc. Furthermore, these factors mostly comprised compounds with $C \leq 15$ (Figure S10b), as expected for a precursor composition of $C_{15}H_{24}$. Due to the prevalence of acyclic structures in the $C_{15}$ carbon skeletons of both farnesene and bisabolene (in particular exocyclic double bonds), the investigated SQTmix is more prone to undergo fragmentation, compared with those sesquiterpenes dominated by cyclic structures (e.g. β-caryophyllene) (Faiola et al., 2019). As these smaller fragments can undergo oligomerization reaction, compounds with $C < 15$ can also be oligomers (e.g., a $C_{14}$ compound as

combination of two $C_7$ fragments). However, elucidating the detailed formation mechanisms of the observed compounds in SQTmix SOA particles goes beyond the scope of this study.

Like the AF5 factor in the α-pinene SOA case, the SF5 factor in the SQTmix SOA case contained mainly small compounds with MW < 200 Da despite displaying a continuous increase in signals at temperature above 100 °C (Figure 4). This, again, suggests that thermal decomposition was the main source process when compounds of SF5 were being desorbed from the FIGAERO filter.
Consistently, the compositional profile of SF5 was also dominated by compounds with small carbon numbers (Figure 5b).

### 3.3 Evolution of PMF factors

As shown in the evapogram (Figure 1) and STGs (Figure 2), increasing RH enhanced the evaporation rates of the SOA particles and shifted the particle volatility towards lower C*. These observed changes were caused not only by decreasing particle viscosity (Yli-Juuti et al., 2017; Buchholz et al., 2019; Li et al., 2019) but also possibly by aqueous-phase reactions, especially for highly
oxidized particle samples (Buchholz et al., 2019). To further investigate how particulate water impacts particle evaporation processes here, we need to analyze how the factor volatility and the relative contribution of each factor to the signal of each sample change with isothermal evaporation and humidification. The volatility of each factor can be characterized by its characteristic $T_{desorp}$ values (the 25[th], 50[th] and 75[th] percentile desorption temperature ) of the factor thermogram. The 50[th] percentile is equivalent to $T_{50}$ as used before, while the 25[th] and 75[th] ones indicate the width of a factor thermogram.

Due to different and uncertain amounts of sample mass, it is challenging to investigate changes in the contribution of factors between two evaporation stages by comparing their absolute signals. By normalizing the sum signal of a sample factor k to the total signal (excluding background factors) at the condition j ($F_{k,j}$), we can account for the difference in sample mass. Note that $F_{k,j}$ is not independent of the change in other factors. For instance, if the contribution of the most volatile factor decreases as it is removed by isothermal evaporation faster than other factors, the $F_{k,j}$ values of all other factors will increase. It would not be
possible to separate such behavior from an absolute increase/decrease in the contribution of a factor (e.g., due to a formation/evaporation/decomposition process in the particles) based on the values of $F_{k,j}$ directly. To avoid this issue, we introduce the net change ratio (NCR) using the same rational as for the scaled STG (see section 2.2). We define the NCR as the ratio between the relative contribution of a sample factor $k$ at a given condition $j$ ($F_{k,j}$) and that at the reference condition ($F_{k,ref}$) scaled by the changes caused by the overall evaporation of the particles:

$$NCR_{k,j} = \frac{F_{k,j}}{F_{k,ref}} \cdot \frac{VFR_{avg,j}}{VFR_{avg,ref}} \cdot \alpha_{MW_{avg,j}}^{-1} \cdot \beta_{\rho_{avg,j}} \tag{5}$$

where $F_{k,j}$ and $F_{k,ref}$ are the contributions of a sample factor k to the total signal (excluding background factors) measured by FIGAERO-CIMS at the condition $j$ and reference condition, respectively. $VFR_{avg,j}$ and $VFR_{avg,ref}$ are the mean values of VFR retrieved from SMPS measurements at the condition $j$ and reference condition. $a_{MW_{avg,j}}$ and $\beta_{\rho_{avg,j}}$ are similar to the $a_{MW_{avg}}$ and $\beta_{\rho_{avg}}$ parameters used in Eq. (4). It is not possible to capture the true initial state of particles, as particles start to evaporate directly
after size selection. The dry and fresh condition exhibited the least amount of isothermal evaporation and thus was chosen as the reference case. More details about the derivation of Eq. (5) and the estimation of the parameters can be found in the Appendices B and C, respectively.

NCR represents the net effect of change in a factor, which is a combination of material loss (i.e., evaporation, chemical reactions) and production (i.e., chemical reactions), at a given condition as compared to the reference condition. If NCR is 1, the loss pathway counterbalances the production one, or no change occurs. NCR values significantly smaller than 1 (taking into account the possible uncertainties and limitations of the methodology, we consider $NCR_{k,j} < \frac{1}{2} NCR_{k,ref}$ being significantly smaller) suggests that the loss pathway outweighs the production one, and vice versa. There are two possible loss pathways: evaporation of compounds or transformation of compounds through chemical reactions. If the NCR is smaller than 1 and simultaneously decreases with increasing isothermal evaporation (i.e., decreasing VFR), it implies that the dominant loss mechanism may be evaporation. On the other hand, complex behavior of the NCR with increasing isothermal evaporation (e.g., a decrease followed by an increase) indicates that the main loss mechanism of the compounds is likely chemical transformation. When NCR is clearly larger than 1 (taking into account the possible uncertainties and limitations of the methodology, we consider $NCR_{k,j} > 2 NCR_{k,ref}$ being significant larger), it implies that the compounds are produced in the particle phase. In addition to the trends in the NCR values, the shape of the factor thermograms and their inferred C* values also give further insights into the possible production and loss mechanisms as discussed below.

### 3.3.1 SQTmix SOA particles

Consistent with the small change in VFR (< 12% in volume), the particle composition in the dry SQTmix SOA particles barely changed (Figure 5, red colors), with negligible shifts only in the NCR of SF1. As seen in Figure 5, for the factors SF1, SF2 and SF4, the NCR decreased with decreasing VFR, implying the contribution of evaporation to the material loss. At high RH, SF1 and SF4 were no longer present after isothermal evaporation in the RTC.

As the range of C* assigned to the characteristic T$_{desorp}$ of SF1 was high enough to enable significant evaporation during the experimental timescale of up to 4.25 h and its NCR exhibited a decreasing trend with evolving evaporation, we can conclude that the decrease in NCR of SF1 was primarily driven by evaporation. In this case, the particulate water mainly accelerated the evaporation as an effective plasticizer. The decrease of NCR for SF2 and SF4, which have volatilities in the LVOC and ELVOC range respectively, was even stronger than that of SF1 at high RH. This was surprising as compounds in that volatility range are not expected to evaporate significantly from particles within 4.5 h at room temperature (Li et al., 2019). Hence this observation indicates that in addition to evaporation, there was another loss mechanism (i.e., aqueous-phase process) driving the evolution of SF2 and SF4 under high RH conditions.

When investigating the factors SF3 and SF5, changes in their NCR were negligible under dry conditions, but significant increases in their NCR were seen at high RH (Figure 5). At the same time, we can see that both of these factors accounted for substantial amounts of the total particle composition at high RH (Figure 4). This clearly indicates that compounds in SF3 or SF5 were not only retained in particle phase due to their low C* values in the range of (E)LVOC, but also formed in the particle phase at high RH. These processes must be relatively fast as the changes in abundance and NCR were already clear at the fresh stage (i.e., within 0.25 h).

Except for SF1, all factors showed a distinct shift to higher values of characteristic T$_{desorp}$ under high RH conditions as compared to dry conditions. This also indicates that the presence of water content has a more complex impact on the particle composition than simply enhancing the isothermal evaporation of volatile compounds. The correlations induced by the aqueous-phase processes are more important than the grouping solely by volatility class. I.e., compounds with a wider range of volatilities may be grouped

into a factor if they are produced by the same chemical process. We provide additional discussion about the possible reasons for the changes of the factor thermogram shapes in the Supplement (see section S1.2.3).

We will further elaborate on the possible reasons for these observed changes in NCR together with those described in the next section for α-pinene SOA particles in section 3.3.3.

### 3.3.2 α-pinene SOA particles

The response of the STG to isothermal evaporation and humidification appeared to be very similar for the α-pinene and SQTmix SOA particles (Figure 2a, b). The investigation of the NCR values of PMF factors revealed that, while the overall behavior was indeed similar, there were also some distinct differences in the chemical composition between these two types of SOA particles.

As expected from the isothermal evaporation measurements and the comparison of the STGs before and after isothermal evaporation in the RTC, the α-pinene SOA particles showed very little change for the NCR under dry conditions (Figure 6, red colors). Under high RH conditions, AF1, AF2, and AF4 exhibited lower NCR values (NCR < 1) compared to the dry conditions (Figure 6). However, a continuous reduction in NCR with decreasing VFR (to the point that no contribution of the factor is detectable) was only observed for AF1. Similar to the case of SF1, we concluded that the evolution of AF1 was primarily driven by the evaporation process controlled by its average $C^*$ which lies in the volatility range between SVOC and LVOC.

The evolution of NCR with decreasing VFR was more complex for AF2 and AF4 as compared with that for AF1: their NCR values did not decrease with decreasing VFR but instead showed an increase with decreasing VFR at high RH. These observations imply that the aqueous-phase chemical transformation were the dominant processes affecting the evolution of AF2 and AF4 at high RH instead of simple evaporation. Such chemical transformations could also cause the increases in the characteristic $T_{desorp}$ and the factor thermogram width observed at high RH (Figure 3a and Figure 6a), in particular for AF2 with its $T_{50}$ increasing from 105 °C to 135 °C.

AF3 exhibited an NCR > 1 in the fresh case under high RH conditions, which means additional amounts of compounds grouped into that factor were formed in the presence of an aqueous phase in the particles. Note that many of the ions grouped into AF3 also showed an increase in the absolute measured signal under high RH conditions after accounting for the different amount of collected sample mass on the filter. With longer isothermal evaporation time, NCR decreased for AF3, which means that some of the compounds grouped into AF3 must have evaporated from the particles or continued to react to form different products grouped into other factors. The change of the factor thermogram shape (i.e. loss of compounds with higher $C^*$ and lower $T_{desorp}$) in Figure 3 together with a minor shift in the characteristic $T_{desorp}$ in Figure 6 suggest that the removal due to evaporation is the more likely explanation. Hence, the evolution of NCR of AF3 at high RH suggests complex behavior including the formation of compounds at the particle phase but also the loss of some compounds mainly by evaporation.

Negligible changes in NCR of AF5 alone indicates minor changes in composition during evaporation under dry or high RH conditions. In addition, when considering that AF5 is (mainly) in the ELVOC range (see Figure 3), the isothermal evaporation of compounds should not be significant in the experimental timescale of up to 4.5h (Li et al., 2019). But when investigating the factor thermograms (Figure 3) in detail, the changes in the shape of the factor thermogram and $T_{desop}$ (Figure 3) together imply that apart from evaporation, water driven aqueous-phase processes also affected at least some of the compounds with extremely low $C^*$ which were grouped into AF5. Although both AF5 and SF5 were dominated by products of thermal decomposition, it does not indicate that their compositions were similar. While the mass spectra of AF5 was dominated by ions with Cnum from 7 to 10,

major ions in the mass spectra of SF5 tended to have Cnum of 6 or below (Figure S10). As these two factors originated from two different SOA systems, it is highly possible that they can behave differently against particulate water. It is also important to remember in this context that the products of any decomposition process may be similar or even identical, but they may stem from completely different parent compounds. Especially, very small fragments (e.g., oxalic acid or acetic acid) carry very little information about the original molecule they came from.

### 3.3.3 Interpretation of the evolution of NCRs

Overall, particulate water not only accelerates the evaporation of sample factors by reducing bulk diffusion limitations, but also alters the chemical composition of particles by inducing chemical aqueous-phase processes (e.g., hydrolysis or oligomerization). Accelerated evaporation primarily driven by the water plasticizing effect was observed for those sample factors with the smallest average MW and the highest volatility in both SOA systems (i.e., α-pinene: AF1 and SQTmix: SF1). On the other hand, changes in the NCR together with changes in the absolute abundance and/or the characteristic $T_{desorp}$ for the other sample factors very likely suggest the presence of aqueous-phase processes that generally modify the composition and volatility of the (remaining) SOA particles.

The factors affected by chemical aqueous-phase processes can be classified as (i) "educt" factors with NCR < 1 and (ii) "product" factors with NCR > 1 under the same conditions. "Educt" factors contain water-labile compounds which are stable under dry conditions but undergo chemical reactions in the presence of water. Likely aqueous-phase reactions are the fragmentation (hydrolysis) of organic (hydro)peroxides (Krapf et al., 2016; Zhao et al., 2018; Qiu et al., 2019) or accretion reactions. Examples for these "educt" factors were SF2, SF4, AF2, and AF4. All these factors exhibited NCR values clearly < 1, while their volatilities were in the (E)LVOC range which makes a substantial isothermal evaporation within 0.25 h very unlikely.

The products of these aqueous-phase reactions will evaporate from the particle phase if their volatility is high enough (e.g., small fragments from fragmentation reactions). Products with sufficiently low volatility will remain in the particle phase and contribute to the "product" factors. Such compounds with sufficiently low volatility may be the larger fragments of fragmentation reactions, but the majority is likely formed from accretion reactions such as (i) non-oxidate reactions involving two or more carbonyls (i.e., (hemi)acetal formation, aldol condensation, and esterification), or (ii) reactions incorporating carbonyls and organic hydroperoxides (i.e., peroxy(hemi)acetal formation) (Kroll and Seinfeld, 2008; Herrmann et al., 2015). The predominant non-oxidative nature of these reactions is dictated by the fact that the average OSc of the particles does not increase under high RH conditions.

The "product" factors for the SQTmix SOA particles (SF3 and SF5) were also identifiable by the fact that they have almost no contribution to the total signal under dry conditions. The comparable "product" factor for the α-pinene SOA particles (AF3) already contributed to the particles under dry conditions and then showed an increase in contribution under high RH conditions. This behavior is probably linked to the SOA production inside the OFR which was at ~ 40% RH. For α-pinene SOA, compounds grouped into AF3 could be already produced inside the OFR either in the gas phase or by the particle-phase processes. The absence/very small contribution of SF3 or SF5 under dry conditions indicates that the processes leading to their formation were too slow to produce significant amounts during the short residence time prior to the particle size selection.

Another difference between the two SOA types lies in the evolution of the "educt" and "product" factors in the RTC under high RH conditions. For the SQTmix SOA particles, the evolution of the NCR values of all factors was monotonic (i.e., either increasing

or decreasing with decreasing VFR). This may indicate that the underlying dominant process is either a removal or a production process for each factor. It should be noted that multiple loss and production processes may coexist for a factor, especially at high RH where aqueous-phase processes may play a role. For instance, the removal of compounds grouped into the "educt" factor AF2 or AF4 via chemical reactions was dominant over any production process. But with increasing isothermal evaporation time at high RH, the balance between these processes shifted slightly, leading to a small increase in the NCR. The balance between the removal and production of compounds may vary over time. This is probably the cause of the complex behavior of NCR values for AF2, AF3 and AF4 and may be coupled to the observed changes in the factor thermogram shapes for these factors.

Although there are multiple studies of isothermal evaporation of α-pinene SOA particles, very few studies provide molecular information that is comparable to our approach. D'Ambro et al. (2018) conducted FIGAERO-CIMS measurements of particles that evaporated on the filter after collection. Although the α-pinene SOA particles in the study may not be directly comparable to the particles in our study, some of their findings share similarities with the interpretation of our PMF factors. For each ion, they explain the observed isothermal evaporation behavior with a model containing 3 components with different apparent volatility: (i) free monomers that evaporate from particles according to their $C^*$ values, (ii) ELVOC compounds that do not evaporate from the particles at room temperature but decompose upon heating to be detected as the single ion, and (iii) reversible oligomers that decompose into the corresponding free monomers with time or heat. In our data set, many individual ions show contributions from multiple factors. AF1 and SF1 are predominantly containing compounds that behave like "free monomers". AF5 and SF5 are mostly ELVOC compounds that are detected as thermal decomposition products. The behavior described for "reversible oligomers" is in line with the complex behavior of the PMF factors which we associate with aqueous-phase processes. As D'Ambro et al. (2018) only applied their model investigation to particle evaporation at 50% RH and above, it is impossible to determine whether the particle phase processes affecting the reversible oligomers are linked to the presence of particulate water. Note that the approach of D'Ambro et al. (2018) deploys a ion-by-ion model fitting, while our PMF analysis inspects the behavior of all ions in the data set at once.

## 4. Atmospheric implications and conclusions

This isothermal evaporation study demonstrates that the SQTmix SOA particles evaporated slower than the α-pinene ones. Additional compositional measurements with FIGAERO-CIMS enabled the separation of particulate constituents by their volatilities. By examining the particle samples at two different evaporation stages (fresh vs. RTC), we observed relatively less changes in $T_{50}$ and smaller decreases in the STGs of the SQTmix SOA particles, in comparison to the α-pinene SOA particles. This is in line with the observation of slower evaporation rates of the SQTmix SOA particles during isothermal evaporation. Compared to the α-pinene SOA particles generated under comparable oxidation conditions, the overall less evaporation of the SQTmix SOA particles can be attributed to its higher OSc value which is consistent with its lower volatility and possibly higher viscosity.

To our knowledge, this is the first study investigating the volatility of SOA particles from a mixture of farnesene and bisabolene which are acyclic and monocyclic sesquiterpenes of atmospheric relevance. For α-pinene, multiple studies of isothermal evaporation at room temperature exist (Vaden et al., 2011; Wilson et al., 2014; Yli-Juuti et al., 2017; D'Ambro et al., 2018; Buchholz et al., 2019; Li et al., 2019; Zaveri et al., 2020; Pospisilova et al., 2021). However, even for this single precursor system, the formation conditions determine the isothermal evaporation behavior of the formed SOA and thus must be carefully considered when comparing different studies. The detailed composition of particles determines their volatility, viscosity, and behavior towards

particulate water. Generally, particles containing increasing amounts of higher oxidized compounds will exhibit lower volatility (Buchholz et al., 2019; Zaveri et al., 2020; Pospisilova et al., 2021), but may be more likely to be susceptible to aqueous-phase reactions (Buchholz et al., 2019). Unfortunately, not all previous studies provide an O:C, OSc value or similar proxy to estimate the degree of oxidation, which makes further comparisons difficult.

As the monoterpene with the largest emissions globally (Guenther et al., 2012), α-pinene has commonly served as a model precursor to generate biogenic SOA particles for laboratory studies. Results from these studies have been used to represent properties of many other terpene-derived SOA particles (excluding isoprene-derived SOA) in aerosol-climate models (O'Donnell et al., 2011; Gordon et al., 2016). Our study corroborates previous findings that sesquiterpene-derived particles are more viscous (Saukko et al., 2012), less hygroscopic (Pajunoja et al., 2015), and less volatile (Ylisirniö et al., 2020), compared to α-pinene SOA particles. Since the interplay of particle viscosity and volatility does impact the evaporation dynamics of particles, future studies should focus on particles derived from terpene precursors other than α-pinene to provide better parameterization to comprehensively constrain the gas -particle partitioning behavior of different biogenic SOA particles.

We applied PMF to deconvolute the FIGAERO-CIMS data by grouping desorbed organic compounds into several sample factors. Compared to the full mass spectra, such  statistical analysis provides a useful simplification for describing how particle composition evolves during isothermal evaporation. In line with the minor change in the VFR under dry conditions, there was little difference in the particulate composition between the fresh and RTC samples. On the other hand, the presence of particulate water dramatically altered the dry particle composition at high RH, likely by acting both as a plasticizer for bulk-surface diffusion and a catalyzer for aqueous-phase processes. In each SOA system, the most volatile factor was primarily lost via evaporation when high content of particulate water was present. As suggested by the change in NCR, the water-driven aqueous processes mainly governed the production and/or removal of other sample factors at high RH. Depending on the particle type, sample factors, and evaporation timescale, the effect of aqueous processes could be net production or net loss, which is indicated by the coevolution of particle VFR and factor NCR. While each sample factor of the SQTmix SOA particles was largely controlled by a single type of process, the factors of the α-pinene ones evolved according to the complex and time-dependent interplay of production and removal processes.

The observed aqueous-phase processes are not unique to SOA particles formed in the OFR. Prevalence of ether groups has been observed in ambient particles with high aerosol liquid water content, suggesting abundant formation of (hemi)acetals from carbonyls (Gilardoni et al., 2016; Ditto et al., 2020). Additionally, the prevalence of terpene-derived oligomers as well as carbon chain lengths have been found to decrease in cloud-water samples as compared to particle samples collected below cloud, indicating the possible presence of hydrolysis in cloud water (Boone et al., 2015). Although increasing evidence from laboratory and field observations have suggested the importance of aqueous-phase processes, such reactions are still underrepresented in the existing models because of a lack of fundamental knowledge (McNeill, 2015). While the aqueous-phase processes of simple, typically small carbonyl compounds have been well studied so far (De Haan et al., 2009; Schwier et al., 2010; Yasmeen et al., 2010; Li et al., 2011; Zhao et al., 2012; Zhao et al., 2013; Petters et al., 2020), more studies should investigate the processes involving complex and large molecules with multiple functional groups.

## Appendix A. Scaled sum thermograms of RTC samples

To investigate the changes in volatility of SOA particles, we need to compare the number of ions at each desorption temperature between fresh (0.25 h, $N_{fresh}(T)$) and RTC (4.25 h, $N_{RTC}(T)$) samples collected on the FIGAERO filter. The remaining fraction of all ions (RF) observed at a given temperature in each sample can be described as:

$$RF_{fresh}(T) = \frac{N_{fresh}(T)}{N_{0,fresh}(T)} \tag{A1}$$

$$RF_{RTC}(T) = \frac{N_{RTC}(T)}{N_{0,RTC}(T)} \tag{A2}$$

where $N_{0,fresh}(T)$ and $N_{0,RTC}(T)$ are the number of ions at each desorption temperature at each initial stage, i.e., before any isothermal evaporation occurred for the fresh and RTC samples, respectively. Note that $N_{0,fresh}(T)$ and $N_{0,RTC}(T)$ depend on the collected sample amount in each case.

The total remaining fraction of ions across the whole range of desorption temperatures ($RF_{Tot}$) is equal to:

$$RF_{Tot,fresh} = \frac{\sum_0^T N_{fresh}(T)}{\sum_0^T N_{0,fresh}(T)} = \frac{N_{Tot,fresh}}{N_{Tot,0,fresh}} \tag{A3}$$

$$RF_{Tot,RTC} = \frac{\sum_0^T N_{RTC}(T)}{\sum_0^T N_{0,RTC}(T)} = \frac{N_{Tot,RTC}}{N_{Tot,0,RTC}} \tag{A4}$$

where $N_{Tot,fresh}$ and $N_{Tot,RTC}$ are the sum of all ions over all desorption temperatures at the fresh and RTC stages. $N_{Tot,0,fresh}$ and $N_{Tot,0,RTC}$ are the same sums but at the initial stage before any isothermal evaporation occurred for each sample.

In the absence of a reliable sensitivity calibration of the CIMS, the measured STG at a given desorption temperature ($STG_{fresh}(T)$ and $STG_{RTC}(T)$) is equivalent to the number of ions detected at this desorption temperature ($N_{fresh}(T)$ and $N_{RTC}(T)$). To account for different amounts of mass loadings on the FIGAERO filter, we normalize the measured STG with the total ion signal of each sample ($N_{Tot}$):

$$STG_{N,fresh}(T) = \frac{N_{fresh}(T)}{N_{Tot,fresh}} \tag{A5}$$

$$STG_{N,RTC}(T) = \frac{N_{RTC}(T)}{N_{Tot,RTC}} \tag{A6}$$

With Eq. (A1) − (A4), the expressions for the normalized STGs in Eq. (A5) and (A6) can be converted to:

$$STG_{N,fresh}(T) = \frac{N_{0,fresh}(T) \cdot RF_{fresh}(T)}{N_{Tot,0,fresh} \cdot RF_{Tot,fresh}} \tag{A7}$$

$$STG_{N,RTC}(T) = \frac{N_{0,RTC}(T) \cdot RF_{RTC}(T)}{N_{Tot,0,RTC} \cdot RF_{Tot,RTC}} \tag{A8}$$

Due to experimental limitations, different amounts of sample were collected in the fresh and RTC cases. Thus, the total signal at the corresponding initial stage is not equal either. However, the ratio between $N_0(T)$ and $N_{Tot,0}$ is independent of the amount of sample and can be expressed as:

$$k(T) = \frac{N_{0,fresh}(T)}{N_{Tot,0,fresh}} = \frac{N_{0,RTC}(T)}{N_{Tot,0,RTC}} \tag{A9}$$

Comparing the normalized STG is not equivalent to the direct comparison between $RF_{fresh}(T)$ and $RF_{RTC}(T)$, since $RF_{Tot,fresh}$ and $RF_{Tot,RTC}$ are not equal. We assume that the change in $RF_{Tot}$ is determined by the isothermal evaporation, which is proportional to the change in the mean value of volume fraction remaining (VFR$_{avg}$). The VFR$_{avg}$ from the isothermal evaporation experiment must be converted to the molar scale first:

$$VFR_{avg,fresh} = \frac{N_{Tot,0,fresh} \cdot RF_{Tot,fresh}}{N_{Tot,0,fresh}} \cdot \frac{MW_{avg,fresh}}{MW_{avg,0}} \cdot \frac{\rho_{avg,0}}{\rho_{avg,fresh}}$$

$$= RF_{Tot,fresh} \cdot \frac{MW_{avg,fresh}}{MW_{avg,0}} \cdot \frac{\rho_{avg,0}}{\rho_{avg,fresh}} \tag{A10}$$

$$VFR_{avg,RTC} = \frac{N_{Tot,0,RTC} \cdot RF_{T,RTC}}{N_{Tot,0,RTC}} \cdot \frac{MW_{avg,RTC}}{MW_{avg,0}} \cdot \frac{\rho_{avg,0}}{\rho_{avg,RTC}}$$

$$= RF_{Tot,RTC} \cdot \frac{MW_{avg,RTC}}{MW_{avg,0}} \cdot \frac{\rho_{avg,0}}{\rho_{avg,RTC}} \tag{A11}$$

where $MW_{avg,fresh}$, $MW_{avg,RTC}$, and $MW_{avg,0}$ are the average molecular weight of the organic compounds and $\rho_{avg,fresh}$, $\rho_{avg,RTC}$, and $\rho_{avg,0}$ are the average particle density for the fresh, RTC, and initial stage, respectively.

Using Eq. (A10) and Eq. (A11), we can express the change in VFR$_{avg}$ between fresh and RTC samples as:

$$\frac{VFR_{avg,RTC}}{VFR_{avg,fresh}} = \frac{RF_{Tot,RTC}}{RF_{Tot,fresh}} \cdot \frac{MW_{avg,RTC}}{MW_{avg,fresh}} \cdot \frac{\rho_{avg,fresh}}{\rho_{avg,RTC}} \tag{A12}$$

Changes in the average molecular weight ($MW_{avg}$) of organic compounds and the average particle density ($\rho_{avg}$) during the isothermal evaporation can be expressed using $a_{MW_{avg}}$ and $\beta_{\rho_{avg}}$:

$$a_{MW_{avg}} = \frac{MW_{avg,RTC}}{MW_{avg,fresh}} \tag{A13}$$

$$\beta_{\rho_{avg}} = \frac{\rho_{avg,RTC}}{\rho_{avg,fresh}} \tag{A14}$$

Rearranging Eq. (A12) and using the definitions in Eq. (A13) and Eq. (A14), we can express the change in $RF_{Tot}$ with the removal factor, $f_{removal}$:

$$f_{removal} = \frac{RF_{Tot,RTC}}{RF_{Tot,fresh}} = \frac{VFR_{avg,RTC}}{VFR_{avg,fresh}} \cdot \alpha_{MW_{avg}}^{-1} \cdot \beta_{\rho_{avg}} \tag{A15, Eq. (3) in main text}$$

To remove the term $RF_{Tot,RTC}$ in Eq. (A8), we multiple Eq. (A8) with Eq. (A15) to calculate the scaled STG ($STG_{SC,RTC}(T)$).

$$STG_{SC,RTC}(T) = STG_{N,RTC}(T) \cdot f_{removal} = \frac{N_{0,RTC}(T) \cdot RF_{RTC}(T)}{N_{Tot,0,RTC} \cdot RF_{Tot,fresh}} \tag{A16}$$

Eq. (A16) can be also expressed in the form in Eq. (A17) which is equivalent to Eq. (4) in the main text.

$$STG_{SC,RTC}(T) = STG_{N,RTC}(T) \cdot \frac{VFR_{avg,RTC}}{VFR_{avg,fresh}} \cdot \alpha_{MW_{avg}}^{-1} \cdot \beta_{\rho_{avg}} \tag{A17, Eq. (4) in main text}$$

Now we can rearrange Eq. (A7) and Eq. (A16) as follows:

$$RF_{fresh}(T) = \frac{N_{Tot,0,fresh} \cdot RF_{Tot,fresh}}{N_{0,fresh}(T)} \cdot STG_{N,fresh}(T) \tag{A18}$$

$$RF_{RTC}(T) = \frac{N_{Tot,0,RTC} \cdot RF_{Tot,fresh}}{N_{0,RTC}(T)} \cdot STG_{SC,RTC}(T) \tag{A19}$$

Using Eq. (A9), these can be simplified as:

$$RF_{fresh}(T) = \frac{RF_{Tot,fresh}}{k(T)} \cdot STG_{N,fresh}(T) \tag{A20}$$

$$RF_{RTC}(T) = \frac{RF_{Tot,fresh}}{k(T)} \cdot STG_{SC,RTC}(T) \tag{A21}$$

These two equations show that comparing $STG_{N,fresh}(T)$ with $STG_{SC,RTC}(T)$ is equivalent to the direct comparison between $RF_{fresh}(T)$ and $RF_{RTC}(T)$.

### Appendix B. Calculation of net change ratio (NCR) for each PMF sample factor

We want to investigate the evolution of each sample factor $k$ during isothermal evaporation by comparing its contribution to the total particle composition at different conditions $j$ (fresh vs RTC; dry vs. high RH). To account for different amounts of collected sample material on the FIGAERO filter, we normalize the measured sum of ions from a factor $k$ ($N_{k,j}$) to the total ion signal of each sample ($N_{Tot,j} = \sum_{k=1}^{5} N_{k,j}$) excluding the contribution of background factors. The contribution of a factor $k$ ($F_{k,j}$) to each sample can be calculated as:

$$F_{k,j} = \frac{N_{k,j}}{\sum_{k=1}^{5} N_{k,j}} = \frac{N_{k,j}}{N_{Tot,j}} \tag{B1}$$

The remaining fraction of a sample ($RF_{Tot,j}$) can be calculated as follows:

$$RF_{Tot,j} = \frac{N_{Tot,j}}{N_{Tot,0,j}} \tag{B2}$$

where $N_{Tot,j}$ is the total ion signal of each sample and $N_{Tot,0,j}$ is the total ion signal at the initial state, i.e., before any isothermal evaporation occurred for the collected sample. It should be noted that the value of $N_{Tot,0,j}$ depends on the collected mass at each

560 condition j.

In the same manner, the remaining fraction of a sample factor $k$ at a condition $j$ ($RF_{k,j}$) can be defined as:

$$RF_{k,j} = \frac{N_{k,j}}{N_{k,0,j}} \tag{B3}$$

where $N_{k,0,j}$ is the total ion signal of a factor k at the initial state. Similar to $N_{Tot,0,j}$, the value of $N_{k,0,j}$ also depends on the total sum signal of a sample $k$ at a condition $j$.

Expressing $N_{Tot,j}$ and $N_{k,j}$ in Eq. (B1) with Eq. (B2) and Eq. (B3) yields:

$$F_{k,j} = \frac{N_{k,j}}{N_{Tot,0,j} \cdot RF_{T,j}} = \frac{N_{k,0,j} \cdot RF_{k,j}}{N_{Tot,0,j} \cdot RF_{Tot,j}} \tag{B4}$$

In the same manner as $\frac{N_0(T)}{N_{Tot,0}}$ expressed in Eq. (A9), the ratio between $N_{k,0}$ and $N_{Tot}$ is also independent of the amount of sample

$$\frac{N_{k,0,j}}{N_{Tot,0,j}} = \frac{N_{k,0,ref}}{N_{Tot,0,ref}} \tag{B5}$$

It is not possible to capture the true initial state of particles, as particles start to evaporate directly after size selection. The dry and fresh condition exhibited the least amount of isothermal evaporation and thus was chosen as the reference case. By comparing $RF_{k,j}$ of the other sample with $RF_{k,ref}$, we could gain insights into the effect of increasing evaporation time and/or RH on each sample factor $k$. Here, we introduce the net change ratio (NCR) which is defined as the ratio between the remaining fraction of a sample factor $k$ at a condition $j$ ($RF_{k,j}$) and that at the reference condition ($RF_{k,ref}$). The principle of NCR is comparable to the scaling treatment applied to the STG(T) of RTC samples (Appendix A). The NCR for a sample factor $k$ at a condition $j$ ($NCR_{k,j}$) can be expressed as follows:

$$NCR_{k,j} = \frac{RF_{k,j}}{RF_{k,ref}} \tag{B6}$$

Using Eq. (B4) and Eq. (B5), we rearrange Eq. (B6) and present $NCR_{k,j}$ as follows:

$$NCR_{k,j} = \frac{F_{k,j}}{F_{k,ref}} \cdot \frac{RF_{Tot,j}}{RF_{Tot,ref}} \tag{B7}$$

Note that the value of $NCR_{k,j}$ is not equivalent to the ratio of contribution of a factor $k$ between the condition $j$ and reference condition, since $RF_{Tot,j}$ is not equal to $RF_{Tot,ref}$. Similar to the scaled STG approach, the change in $RF_{Tot}$ is assumed to be proportional to that in the VFR between two conditions (the condition $j$ vs the reference condition). Similar to Eq. (A12), the ratio of $RF_{Tot}$ between a condition $j$ and reference condition can be solved as

$$\frac{RF_{Tot,j}}{RF_{Tot,ref}} = \frac{VFR_{avg,j}}{VFR_{avg,ref}} \cdot \frac{MW_{avg,ref}}{MW_{avg,j}} \cdot \frac{\rho_{avg,j}}{\rho_{avg,ref}} = \frac{VFR_{avg,j}}{VFR_{avg,ref}} \cdot \alpha^{-1}_{MW_{avg,j}} \cdot \beta_{\rho_{avg,j}} \tag{B8}$$

where $\alpha_{MW_{avg,j}}$ and $\beta_{\rho_{avg,j}}$ capture changes in signal-weighted molecular weight ($\frac{MW_{avg,j}}{MW_{avg,ref}}$) and particle density ($\frac{\rho_{avg,j}}{\rho_{avg,ref}}$) between a condition j and reference condition, respectively.

We replace $\frac{RF_{Tot,j}}{RF_{Tot,ref}}$ in Eq. (B7) with Eq. (B8) and then the $NCR_{k,j}$ of a factor $k$ at a condition $j$ can be expressed with the following equation:

$$NCR_{k,j} = \frac{F_{k,j}}{F_{k,ref}} \cdot \frac{VFR_{avg,j}}{VFR_{avg,ref}} \cdot \alpha^{-1}_{MW_{avg,j}} \cdot \beta_{\rho_{avg,j}} \tag{B9, Eq. (5) in main text}$$

**Appendix C. Estimation of average molecular weight ($MW_{avg,j}$) and average particle density ($\rho_{avg,j}$) using PMF sample factors**

For converting the VFR from the volumetric scale to the molar one, values of $MW_{avg,j}$ and $\rho_{avg,j}$ are needed at the condition $j$. For each sample factor $k$ at a condition $j$, we calculate its signal-weighted average molar mass ($MW_{k,j}$) and then estimate its density ($\rho_{k,j}$) using its average O:C and H:C values (Kuwata et al., 2012). For those compounds grouped into factors AF5 and SF5, we are uncertain about the degree of thermal decomposition and that if the decomposition products can be detected by the instrument. In such case, we consider that either none or all of the compounds grouped into these two factors stemmed from thermal decomposition during desorption and we also assume that at least 50% of these thermally labile compounds can be detected by the CIMS. Eventually, we calculate the $MW_{avg,j}$ and $\rho_{avg,j}$ as follows:

$$MW_{avg,j} = \sum_{k=1}^{5} MW_{k,j} \cdot F_{k,j} \tag{C1}$$

$$\rho_{avg,j} = \sum_{k=1}^{5} \rho_{k,j} \cdot F_{k,j} \tag{C2}$$

Using the $MW_{avg,j}$ and $\rho_{avg,j}$ at each condition $j$, we calculate the values of $a_{MW_{avg}}$ and $\beta_{\rho_{avg}}$ used for Eq. (A17) or those of $a_{MW_{avg,j}}$ and $\beta_{\rho_{avg,j}}$ used for Eq. (B9), as summarized in Table C1 and Table C2, respectively. Error bars of these parameters account for the uncertainty arising from the calculation of $MW_{k,j}$ and $\rho_{k,j}$ for factors AF5 and SF5.

**Table C1.** Ranges of parameters for scaling the normalized sum thermograms of RTC stages

| SOA System | Fresh Condition | RTC Condition | $\dfrac{VFR_{avg,RTC}}{VFR_{avg,fresh}}$ | $\alpha_{Mw}$ | $\beta_{\rho_{org}}$ |
|---|---|---|---|---|---|
| α-pinene | Dry, Fresh | Dry, RTC | [0.85, 0.91] | [0.99, 1.01] | [1, 1] |
| | High RH, Fresh | High RH, RTC | [0.57, 0.73] | [1.02, 1.07] | [1.01, 1.01] |
| SQTmix | Dry, Fresh | Dry, RTC | [0.92, 0.95] | [1.01, 1.03] | [1, 1] |
| | High RH, Fresh | High RH, RTC | [0.73, 0.82] | [0.98, 1.03] | [1, 1.01] |

**Table C2.** Ranges of parameters for calculating the net change ratio (NCR) for each PMF sample factor

| SOA System | Ref. Condition | Condition j | $\dfrac{VFR_{avg,j}}{VFR_{avg,ref}}$ | $\alpha_{Mw}$ | $\beta_{\rho_{org}}$ |
|---|---|---|---|---|---|
| α-pinene | Dry, Fresh | Dry, RTC | [0.85, 0.91] | [0.99, 1.01] | [1, 1] |
| | | High RH, Fresh | [0.77, 1.05] | [1.01, 1.07] | [0.99, 0.99] |
| | | High RH, RTC | [0.56, 0.60] | [1.03, 1.14] | [1, 1] |
| SQTmix | Dry, Fresh | Dry, RTC | [0.92, 0.95] | [1.01, 1.03] | [1, 1] |
| | | High RH, Fresh | [0.94, 1.07] | [0.98, 1.33] | [1.01, 1.01] |
| | | High RH, RTC | [0.76, 0.80] | [1.00, 1.31] | [1.01, 1.01] |

*Data availability.* The data set is available upon request from Annele Virtanen (annele.virtanen@uef.fi).

*Supplement.*

*Author contribution.* Z.L., A.B., T.Y.-J and A.V. designed the study. Z.L., A.B., A.Y., L.B., and L.H. carried out laboratory experiments. Z.L., A.B., S.S., T.Y.-J and A.V. performed data analysis and interpretation. Z.L., wrote the paper with contributions from all coauthors.

*Competing interests.* The authors declare that they have no conflict of interest.

*Acknowledgements.* This research was supported by the European Research Council (ERC StG QAPPA 335478), the Academy of Finland Center of Excellence Program (decision 307331), the Academy of Finland (299544, 310682 and 317373), and the University of Eastern Finland Doctoral Program in Environmental Physics, Health and Biology (EPHB).

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

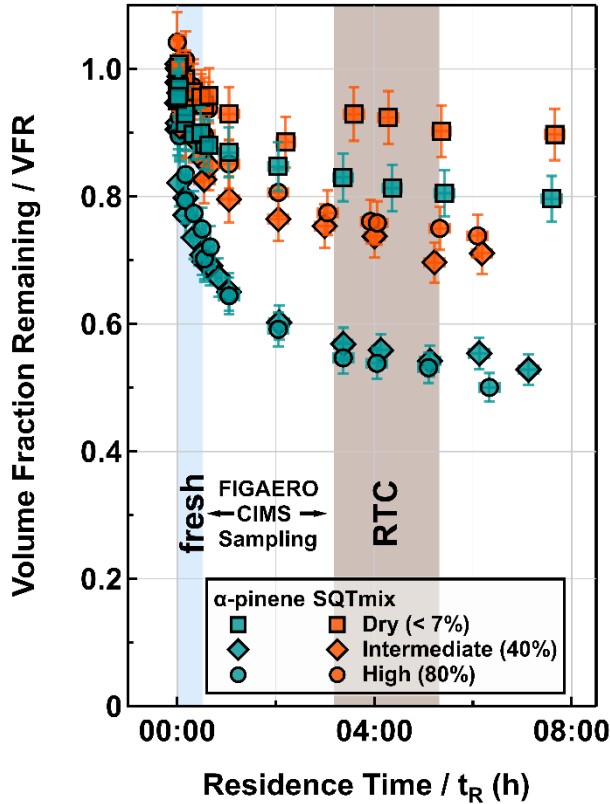

**Figure 1.** Evapograms for α-pinene (turquoise) and SQTmix (orange) SOA particles under dry (<7%), intermediate RH (40% RH) and high RH (80% RH) conditions. The blue (fresh) and brown (RTC) areas indicate the corresponding sampling periods of FIGAERO-CIMS.

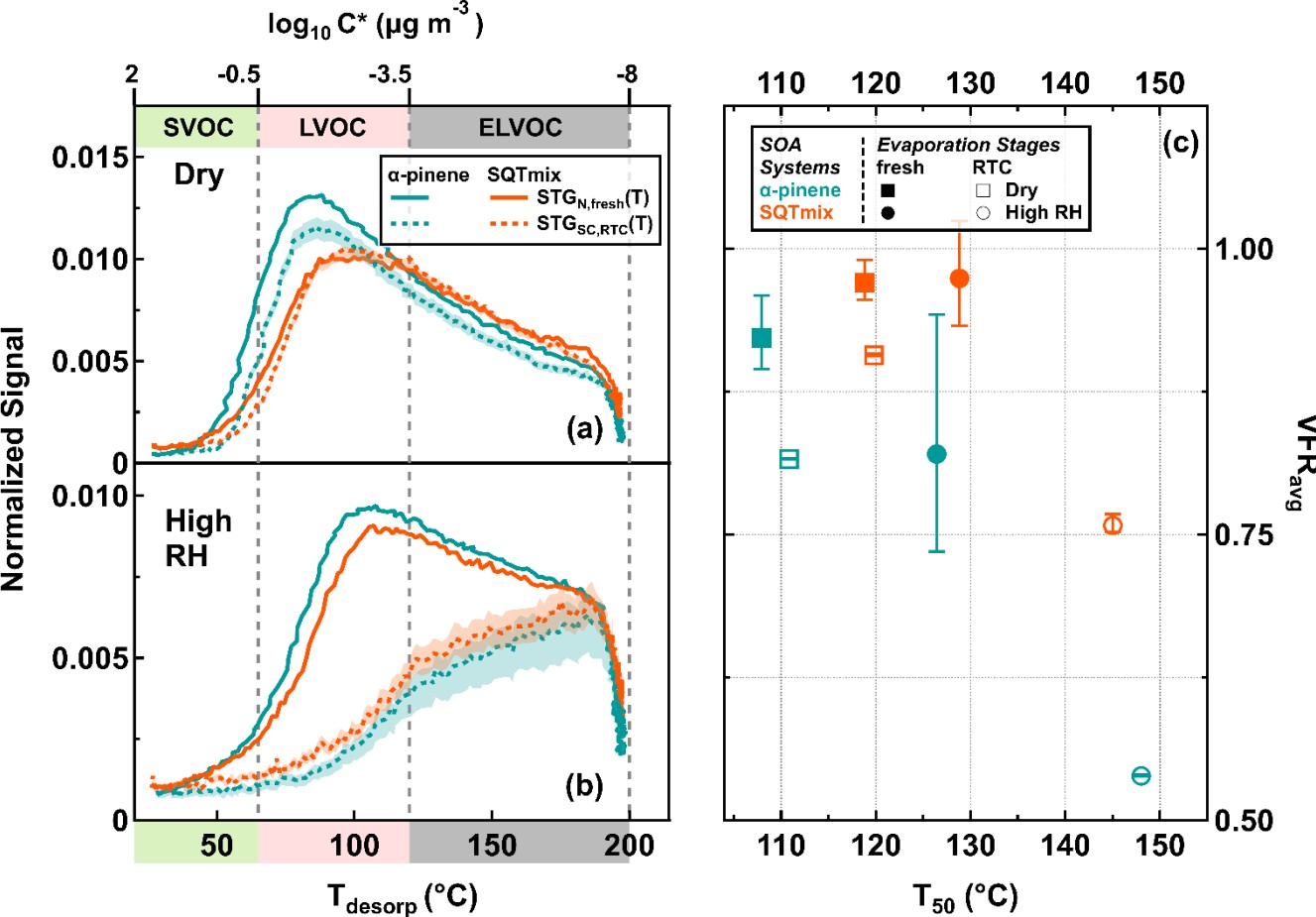

**Figure 2.** Sum thermograms (STG) (a, b), average volume fraction remaining (VFR$_{avg}$) (c), and median desorption temperature (T$_{50}$) (c) for α-pinene (turquoise) and SQTmix (orange) SOA particles, for dry (RH < 7%; (a)) and high RH (RH 80%; (b)) conditions. Shaded areas indicate the ranges of STG(T) for RTC stages after accounting for changes and uncertainties in average molecular weight and particle density (i.e., $\alpha_{MW_{avg}}$ and $\beta_{\rho_{avg}}$ in Eq. (2). Volatility classes (a, b) are derived from T$_{max}$ − C* calibrations using a set of PEG compounds (Ylisirniö et al., 2021). They are indicated by different color bands on the abscissa using the classification according to Donahue et al. (2012).

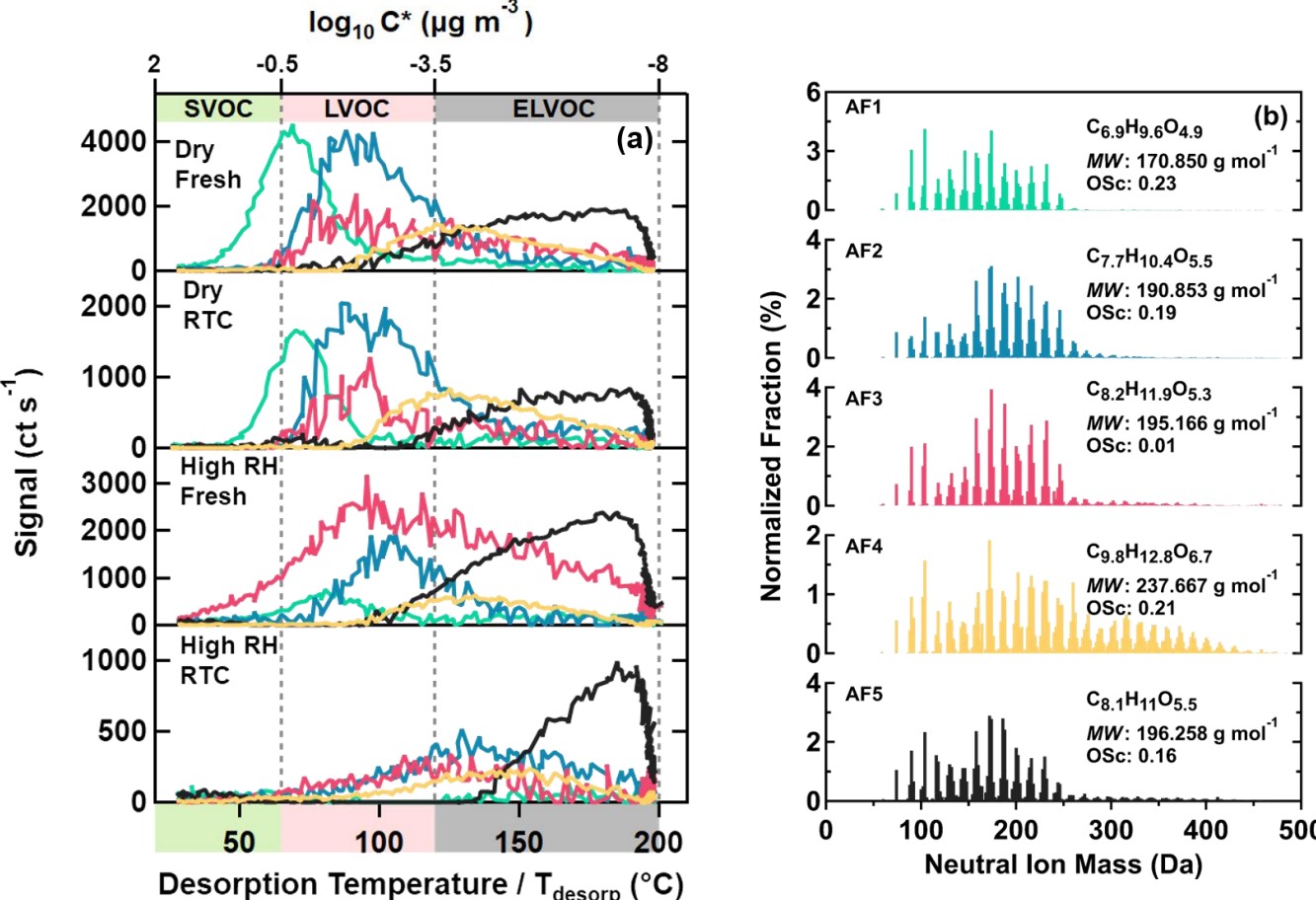

**Figure 3.** Five main sample factors from an eight-factor PMF solution for α-pinene SOA particles. On the panel (a), factor thermograms are shown with color bands on the abscissa indicating volatility classes. On the panel (b), normalized factor mass spectra are presented with their average molecular composition, molecular weight, and oxidation state. The color code is identical for both panels.

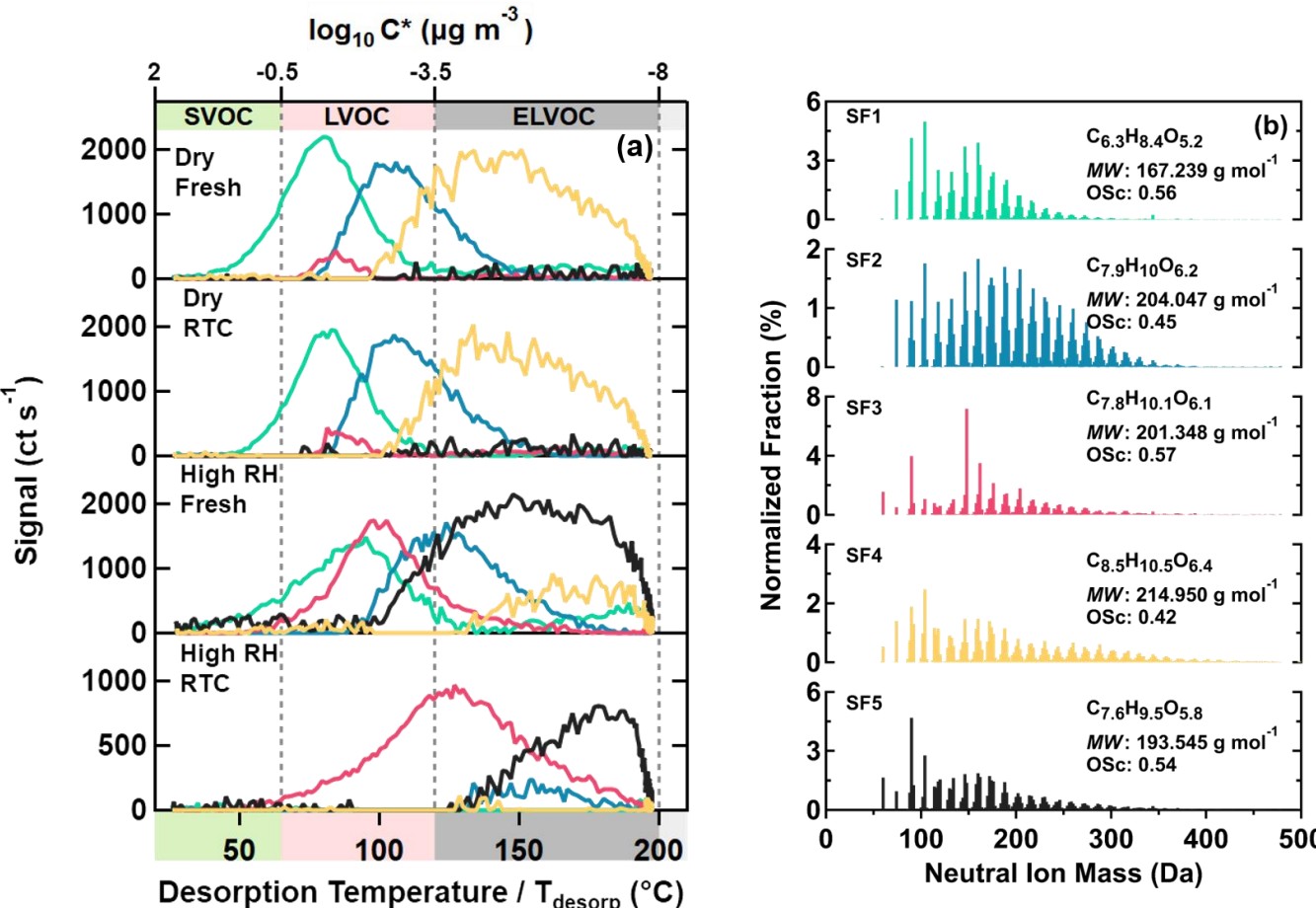

**Figure 4.** Five main sample factors from a ten-factor PMF solution for SQTmix SOA particles. On the panel (a), factor thermograms are shown with color bands on the abscissa indicating volatility classes. On the panel (b), normalized factor mass spectra are presented with their average molecular composition, molecular weight, and oxidation state. The color code is identical for both panels.

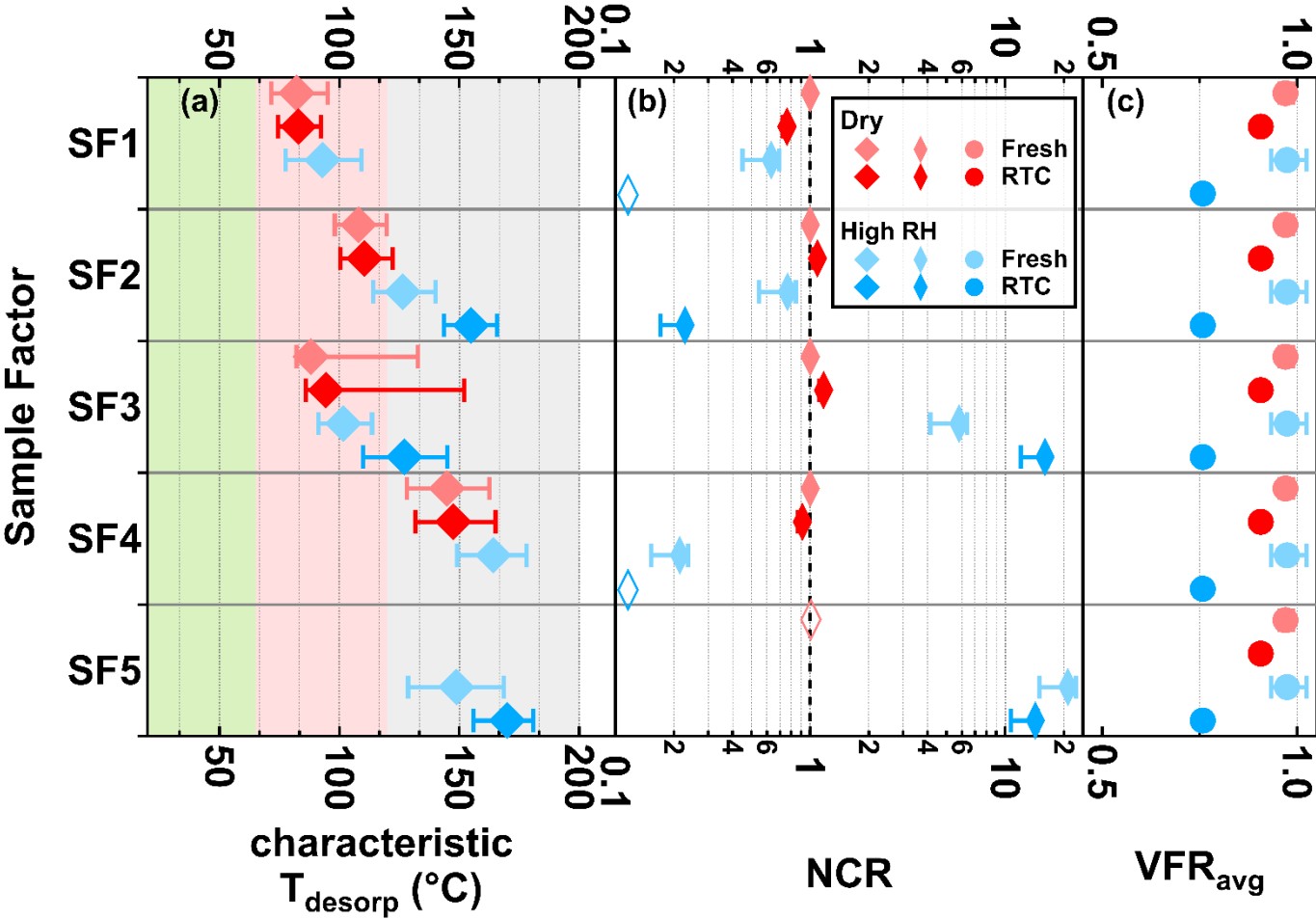

**Figure 5.** Characteristic desorption temperature (characteristic $T_{desorp}$ with 25th, 50th and 75th percentiles, a), net change ratio (NCR, b) of main sample factors and mean values of volume fraction remaining (VFR_avg, c) of the SQTmix SOA particles at fresh (avg. $t_R = 0.25$ h) and RTC (avg. $t_R = 4.25$ h) evaporation stages under dry (red) and high RH (blue) conditions. Background colors in the panel (a) indicate the volatility categories derived from the $T_{max} - C^*$ calibrations (green – SVOC; red – LVOC; and grey – ELVOC). Note that values of VFR_avg are identical in each row of panel (c). The error bars of NCR represent values accounting for changes in molecular weight and particle density, while those of VFR_avg indicates the minimum and maximum values during the FIGAERO sampling time. If the factor thermogram contributes less than 5% to total signals of samples factors and does not exhibit a clear maximum, the corresponding characteristic $T_{desorp}$ values will not be calculated and the NCR will be indicated by an open rhombus close to 0.1.

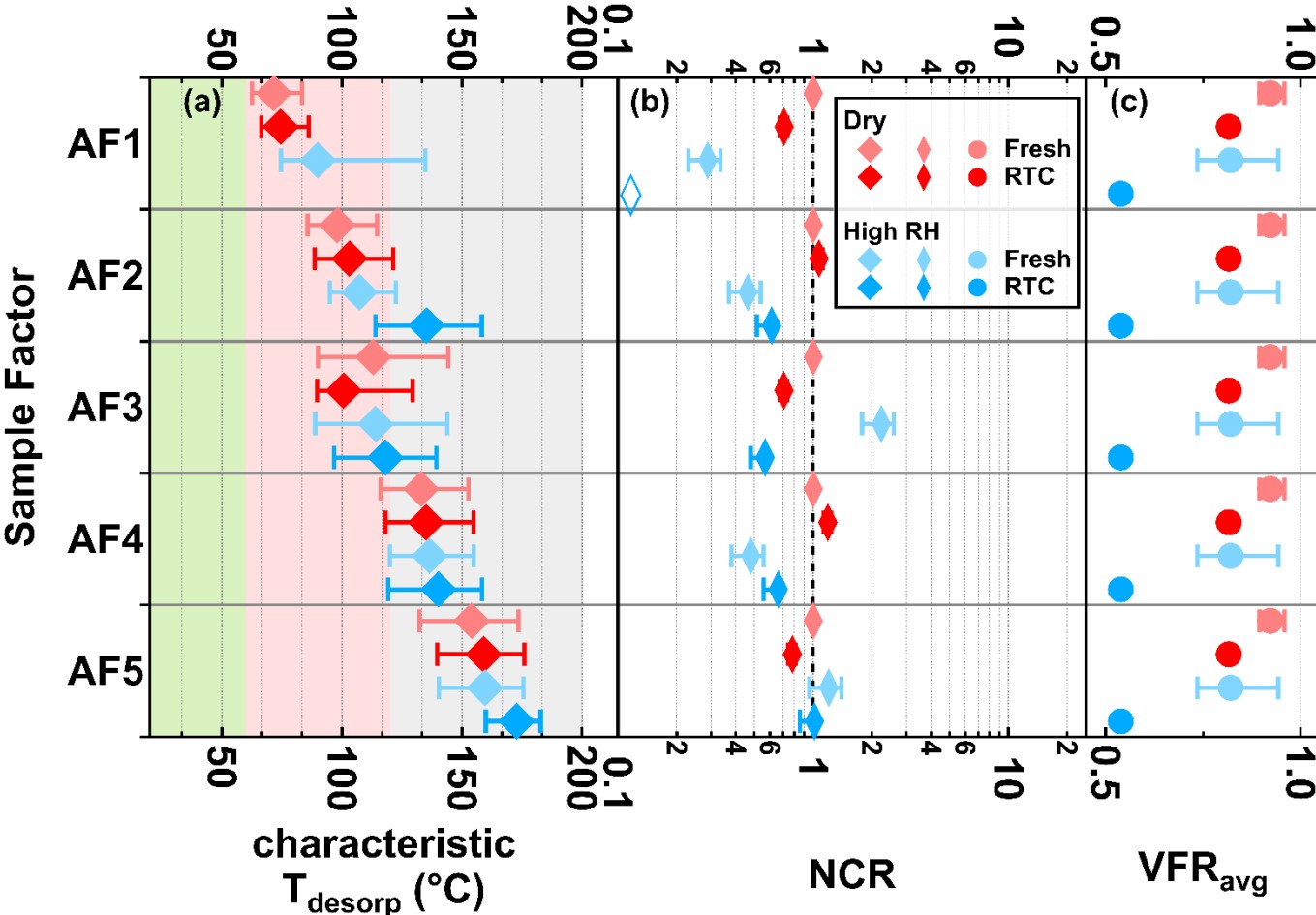

**Figure 6.** Characteristic desorption temperature (characteristic $T_{desorp}$ with 25th, 50th and 75th percentiles, a), net change ratio (NCR, b) of main sample factors and mean values of volume fraction remaining ($VFR_{avg}$, c) of the α-pinene SOA particles at fresh (avg. $t_R = 0.25$ h) and RTC (avg. $t_R = 4.25$ h) evaporation stages under dry (red) and high RH (blue) conditions. Background colors in the panel (a) indicate the volatility categories derived from the $T_{max} - C^*$ calibrations (green – SVOC; red – LVOC; and grey – ELVOC). Note that values of $VFR_{avg}$ are identical in each row of panel (c). The error bars of NCR represent values accounting for changes in molecular weight and particle density, while those of $VFR_{avg}$ indicates the minimum and maximum values during the FIGAERO sampling time. If the factor thermogram contributes less than 5% to total signals of samples factors and does not exhibit a clear maximum, the corresponding characteristic $T_{desorp}$ values will not be calculated and the NCR will be indicated by an open rhombus close to 0.1.

# Evaporation of Sesquiterpene-mixed SOA Particles

## Size Change

## Molecular-level Change

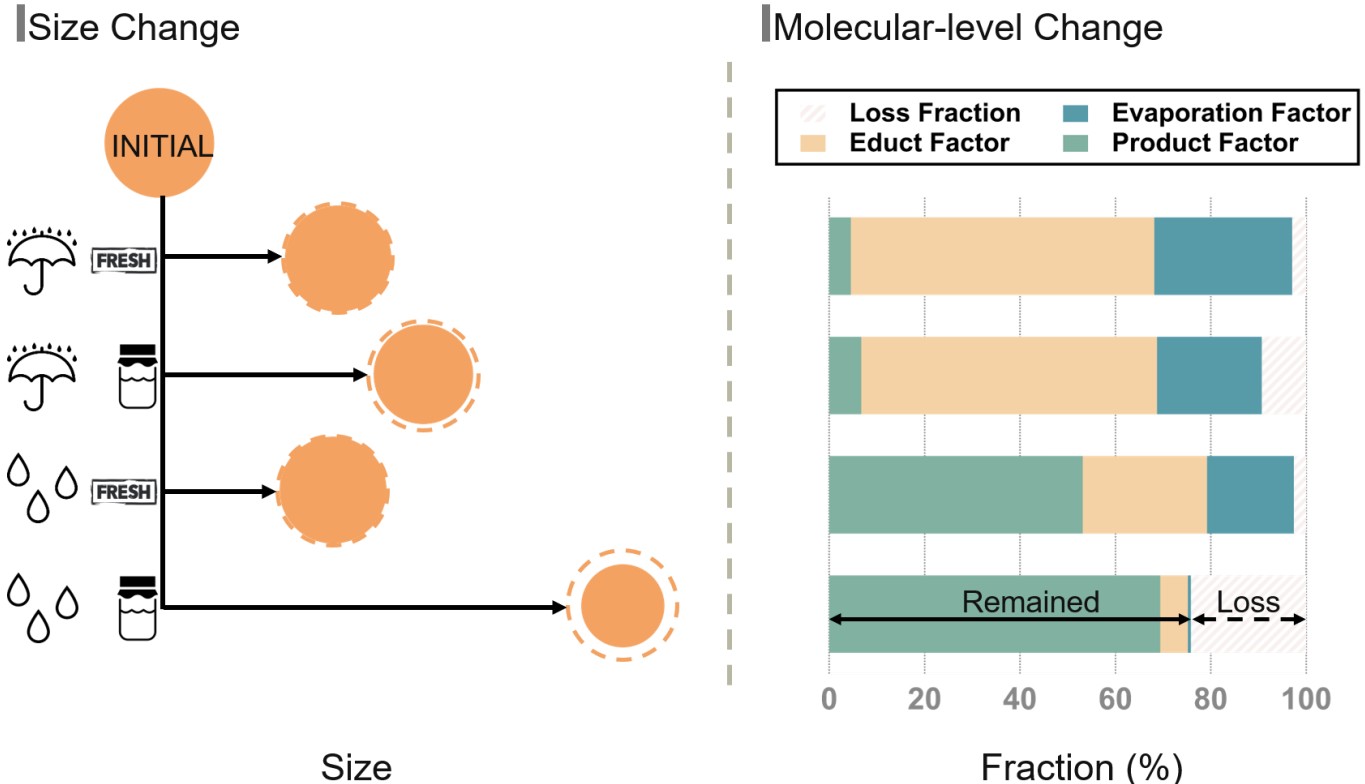

**Figure 7**. For Key Figure Only