# Peer review of "Evolution of volatility and composition in sesquiterpene-mixed and $\alpha$ -pinene secondary organic aerosol particles during isothermal evaporation"

_Atmospheric Chemistry and Physics, 2021_

## Author Comment (AC1)

We gratefully thank all reviewers for the careful reading and valuable comments. Below we provide our point-by-point responses to the reviewer's comments. In the following context, **raised comments/suggestions** are marked in **black**, **responses** are presented in **red**, and **changes to the manuscript/supplement information** are indicated in **blue**. The figures and tables in the following response are numbered consecutively in three replies to reviewers. Additionally, we corrected any minor typo that we recognized in the manuscript and supplement.

The legitimate questions about the real meaning of the PMF factors in the context of thermogram data led us to reconsider the strict distinction of type V and D factors. Instead, we now use the term "sample" factor and relabel the factors as AF1 - AF5 (before AV1-AV4 and AD5) and SF1 - SF5 (before SV1-SV4 and SD5) for α-pinene and SQTmix SOA systems. We use the new labels in our responses to be consistent with the revised manuscript. Note that the interpretation of the factors has not changed, only the labels were adjusted to remove some potential for misunderstandings.

**Reply to Reviewer 1**

Overall, I find this an interesting study that adds to our understanding of particle evaporation behavior and the influence of water on particle composition and evaporation behavior. My major comment relates to the definition of "factors" and how the concept of a factor can be consistent with shifting/evolving evaporation profiles between conditions.

**Specific Comments**

Comment    L170: It is stated that the evaporation rate in dry SOA particles is slowest owing to "considerable kinetic limitations arising from high particle viscosity." As currently written, this is stated as a categorical result. However, there has not yet been discussion of the extent to which chemical differences in the dry versus wet particles could lead to volatility changes separate from viscosity changes. In fact, later in the same paragraph the authors note that aqueous-phase processing occurs when the particles are exposed to 80% RH. I suggest it would be useful if the authors work to better separate results from conclusions to first demonstrate that one effect must be more important than another in determining the overall behavior.

Response    We agree with the reviewer's comment and thus extend the discussion in section 3.1.1.

Changes    Section 3.1.1

[…] The evaporation rate of dry SOA particles is the slowest, and the differences in sum (Figure 2) and factor thermograms (Figure 3 and Figure 4) between two

evaporation stages are minor. The particle evaporation rate became faster with increasing RH for both SOA systems. When particulate water was present, the contribution of compounds in the SVOC range were reduced during fresh stages (Figure 2). As shown in previous studies (Yli-Juuti et al., 2017; Buchholz et al., 2019; Li et al., 2019; Zaveri et al., 2020), considerable kinetic limitations exist for the evaporation of volatile compounds in this type of dry SOA particles due to the substantially high viscosity. Particulate water reduces the viscosity and thus enhances particle evaporation with increasing RH. The comparable evaporation rates under intermediate and high RH conditions suggest that particle evaporation can be approximated assuming liquid-like behavior in this RH range (i.e., RH ≥ 40%). But in addition to this plasticizing effect, particulate water content may also induce chemical aqueous-phase processes during isothermal evaporation (Buchholz et al., 2019; Petters et al., 2020). For the investigated SOA particles, we observed strong evidence of such processes under high RH conditions (RH = 80%). These are detailed in section 3.3.3. Quantifying the effects of particle viscosity and aqueous-phase processes on the SOA particle evaporation would require developing detailed processes models considering particle phase chemistry, which is not the primary focus of this study.

At any set RH, […]

Comment        L208: The authors state that "Each derived factor constitutes a group of organic compounds with the same thermal desorption behavior." While this seems like an a priori true statement, I find it difficult to reconcile with the fact that the authors observe different thermal desorption profiles for the same factors between different conditions. Consider AV2 and AV4 in Fig. 3. Or any of the factors in Fig. 4. The peak desorption temperature and profile shapes change between the different conditions, even at the same RH. This is most evident at high RH. It would seem, therefore, that one factor can have more than one desorption profile and thus different thermal desorption behavior. It would be useful if the authors could provide further discussion regarding such differences for factors. I find it very interesting that the desorption profile for a given factor—presumably, a collection of molecules—should change so much.

Response        We thank the reviewer for directing our attention to this important aspect of our PMF results in this comment and the comment about L 330. It is indeed beneficial to extend the discussion on what a PMF factor represents in the context of FIGAERO thermogram data to help the reader understand the underlying meaning of the observed

changes in the shape of the factor thermogram. We decided to add most of this discussion to section S1.2.3 in the Supplement so that the manuscript is still primarily focused on the interpretation of the data but not shifted too much to the discussion of the analysis method. Additionally, we adjusted the introduction of PMF factors in section 3.2.

There are two types of changes for the factor thermogram shapes: (i) "small" changes in characteristic $T_{desorp}$ of < 15 °C for signals in the S/LVOC range (i.e., AF1 and SF1) and (ii) "large" changes in $T_{50}$ and/or strong changes in the factor thermogram shapes for factors affected by aqueous-phase processes (e.g., AF3, SF3). We will address the "small" changes first and then provide some additional thoughts for the second type. Note that this second type of change is which the reviewer mentioned in the comment about line 330.

The factor mass spectra and factor "time series" (for FIGAERO: factor thermograms) from PMF analysis are indeed "fixed". I.e., the factor mass spectra are identical at each point in time (that means also for each sample), and the normalized factor thermograms are identical for each ion in the same factor. The only variable term is the residual matrix ($E$ in Eq. S2). The interpretation of this mathematical concept is that PMF finds the parts of the signals (ions) that correlate with each other. The reason for that correlation can be a common source or formation process. Specifically, for FIGAERO thermogram data, it can be similar desorption behavior (i.e., similar volatility).

We will use a simplified artificial data set to illustrate the performance of the PMF algorithm. This data set contains 4 "ions" that would be detected by FIGAERO-CIMS. Each ion is constructed using compounds with distinct volatility behavior characterized by their $T_{max}$ and remaining fraction after evaporation. The ions #1 - #3 contain only one compound each (A1, B1, C1, respectively). Ion #4 contains 3 compounds (A2, B2 and C2). The same letter in the compound label means that the compounds exhibit the same volatility. E.g., A1 and A2 both have a $T_{max}$ of 50 °C and show an isothermal evaporation to 0.1 of their starting value.

The thermograms of the individual compounds were constructed using gaussian curves with some random noise.

$$I(T) = 100 * \exp\left(-\left(\frac{T - T_{max}}{5}\right)^2\right) + noise$$

with $I(T)$ intensity of the single compound thermogram, $T_{max}$ peak position of the thermogram and *noise* random noise term.

The ions are formed by using multiples of these compound thermograms ($I_a(T)$, $I_b(T)$, $I_c(T)$) or combinations of them (for ion #4), as summarized in Table R1. Two SOA samples were created: Sample #1 which mimics the fresh case with no evaporation and sample #2 in which the volatile compounds (A1, A2, B1, and B2) have evaporated according to their volatility. The ion thermograms for these samples are shown in Figure R1. Note that the ratios between A2 and B2 are different for the two samples. PMF finds a fairly good solution with 2 factors (see Figure R2 and Table R2): F2 containing C1&2 is identical for sample #1 and #2 as it should be. F1 contains compounds A1&2 and B1&2. The $T_{max}$ value of F1 are 52 and 53 °C for the two samples and the shape of the factor thermogram changes. As the ratio between A and B changes between two samples (i.e., A1&2 are more efficiently removed by isothermal evaporation than B1&2), the smallest residuals are achieved by shifting the thermogram of F1 towards the $T_{max}$ value of B1&2. The PMF reconstruction will not be "perfect" for either sample since neither A1&2 nor B1&2 are captured completely. However, a 2-factor solution is sufficient for interpretating the overall evaporation behavior. We can easily see that F1 is removed with isothermal evaporation and F2 remains. In a 3-factor solution (see Figure R2 and Table R2), A1&2 and B1&2 are separated into two factors F1 and F3 and no change in the thermogram shapes are observed between the samples. For the interpretation, we can state that F1 is a bit more volatile than F3 and both are systematically removed by isothermal evaporation.

**Table R1**. Summary of ions in the artificial data (Table S2 in the Supplement)

| | Compounds | $T_{max}$ / °C | Remaining Fraction | Thermogram – Sample #1 | Thermogram – Sample #2 |
|---|---|---|---|---|---|
| ion #1 | A1 | 50 | 0.1 | $0.75 \times I_a(T)$ | $0.075 \times I_a(T)$ |
| ion #2 | B1 | 55 | 0.5 | $0.75 \times I_b(T)$ | $0.375 \times I_b(T)$ |
| ion #3 | C1 | 70 | 1.0 | $0.5 \times I_c(T)$ | $0.50 \times I_c(T)$ |
| ion #4 | A2, B2, C2 | multi-modal | complex | $1.0 \times I_a(T) + 1.0 \times I_b(T) + 1.0 \times I_c(T)$ | $0.1 \times I_a(T) + 0.5 \times I_b(T) + 1.0 \times I_c(T)$ |

[Figure]

**Figure R1**. Ion thermograms in the artificial data set. (Figure S7 in the supplement)

[Figure]

**Figure R2**. PMF results for artificial data. Factor thermograms (a and b) and residual as time series (c). (Figure S8 in the supplement)

**Table R2**. Compound distributions in PMF factor solutions in the artificial data (Table S3 in the supplement)

| Solution | F1 | F2 | F3 |
|---|---|---|---|
| 2 – factor solution | A1, A2, B1, B2 | C1, C2 | N/A |
| 3 – factor solution | B1, B2 | C1, C2 | A1, A2 |

In a real SOA particle sample, hundreds of compounds covering a range of volatilities are present. One FIGAERO-CIMS ion with a single elemental composition can contain multiple isomers and/or fragments from multiple different parent compounds. These real compounds within one elemental composition may have different desorption behavior (like the A, B, and C compounds in the example above). PMF groups the compounds with the most similar desorption behavior into a few factors with fixed ratios between the ion contributions. The result is always a compromise between finding as few factors as possible and reconstructing the majority of the ion thermograms correctly. For the case of AF1 and SF1, PMF finds the compounds/signals which are most similar (but not identical) in their evaporation behavior. The desorption behavior of the compounds in these factors span over a narrow $T_{max}$-range which we characterize with the characteristic $T_{desorp}$ (e.g., Fig 5a and 6a). During isothermal evaporation, the most volatile fraction of the factor evaporates a little bit more than the rest. Ideally, we would allow adjustments in the factor mass spectrum to account for these changes. However, PMF is not able to do that by design. Instead, it finds a nearly optimal compromise by increasing the characteristic $T_{desorp}$ of the factor to minimize the residuals for all ions present in the factor. Figure R3 shows the behavior of 3 different ions which have a strong contribution to AF1 (factor thermogram shown in red (dry) and blue (high RH)). In the dry and fresh sample (red lines), the single ion thermograms are well reconstructed by PMF (i.e., very small difference between dash vs solid red lines). In the high RH and fresh sample (blue lines), one ion is still reconstructed well (c), one shows an underestimation around $T_{desorp}$ 100 °C (b), and one exhibits an overestimation in that $T_{desorp}$ range (a). This difference in residuals indicates that these 3 ions are experiencing different changes under high RH conditions. I.e., that their "contribution" to the factor mass spectrum has changed somewhat. We carefully inspected the residuals for the reconstructed ion thermograms and found that for each factor with a

significant change in thermogram shape, there were some ions which changed their "goodness of reconstruction" in the relevant $T_{desorp}$ range.

[Figure]

**Figure R3**. Measured (solid lines) and PMF-reconstructed (dashed) ion thermograms. The colored areas indicate the fraction of the signal that is explained by AF1. (Figure S9 in the supplement)

One would assume that this "issue" could be fixed in the same way as for the simple artificial sample above by adding one more factor. We tested the α-pinene data with up to 12 factors and the SQTmix data with 13 factors (i.e., 4 more than in the presented solution). We did not see that the introduction of a factor would separate this slightly different desorption behavior. A splitting into AF1a and AF1b may happen eventually after the occurrence of severe factor splitting and the introduction of multiple "noise" factors. The primary aim of a PMF analysis is to reduce the number of variables for the interpretation step and to identify underlying trends in the data. The behavior of AF1 was interpreted as a systematic removal of compounds mostly affected by isothermal evaporation. A further splitting of AF1 does not provide any additional insight into the overall evaporation behavior of the SOA samples.

For the second type of change in factor thermogram shape (ii), it is necessary to consider aqueous phase processes. The volatility of the factors exhibiting this type of change are in the LVOC or ELVOC range (e.g., AF3). No significant isothermal evaporation is expected for compounds within this range. For AF3, there was a higher

contribution of SVOC compounds in the high RH and fresh case which cannot be explained with any isothermal evaporation pathway. With the available data, we cannot give a definite answer to why these strong changes occur. But we can try to provide a plausible explanation.

The reason for the observed factor thermogram behavior may be connected to the fact that the ions detected in FIGAERO-CIMS are not always the same as the molecules in the particle. One possibility can be thermal induced changes like decarboxylation or dehydration. Another possible option is the thermal decomposition of true dimers (or oligomers) into the monomer units. Assume there are the compounds A, B, C with similar desorption behavior in the sample under dry conditions. They are detected as the ions with the corresponding sum formulas A, B, C. If some of these compounds are "activated" in an aqueous phase to form dimers (e.g., via (hemi-)acetal bonds), there would be the compounds A-B, A-A, A-C, etc. in the sample under high RH conditions. Presumably, the volatility of such dimers is lower than that of the monomer units, but the coupling bond may not be very strong. Very likely, before these compounds can thermally desorb, the bond is already broken and leads to the release of original monomers. The detected ions in FIGAERO-CIMS would again show up as A, B, C with (almost) identical ratios as in the dry case. But $T_{desorp}$ would have shifted to higher values as now we observe thermal decomposition of the dimers instead of direct desorption of the monomers. One can further speculate that with increasing reaction time (i.e., the time in the RTC), the fraction of (low-volatility) oligomers increases which increases the apparent characteristic $T_{desorp}$ of the factor even further.

As the shifts are mostly observed comparing the dry to high RH conditions, the presence of water must play an important role. It could be that the high viscosity in the dry particles also limits the formation processes of such oligomers. The reaction partners have to meet each other which may be very slow in a (semi-)solid phase. But water could also be involved directly in the formation processes. Many of the suggested coupling reactions (e.g., esterification), are catalyzed by available $H^+$ or $OH^-$ in a liquid phase. Similar ideas of reversible oligomer formation have been explored in model calculations (Schobesberger et al., 2018) and were suggested in D'Ambro et al. (2018), Zaveri et al. (2020) and Pospisilova et al. (2021) to explain their observations. Especially for small highly oxygenated ions (e.g., $C_2H_2O_3$), the main

source seems to be from decomposition of different parent compounds at different $T_{desorp}$ values.

Changes       Section 3.2

[...] Each derived factor constitutes a group of organic compounds with very similar "temporal" behavior. The PMF algorithm does not prescribe any meaning to the position of a value in the dataset, i.e., the $T_{desorp}$ or desorption time values are only used to define the order of the data points. When volatility acts as the primary factor driving the composition change in the particles, compounds with similar desorption behavior correlate and are grouped into factors. In each factor, compounds of similar volatility evaporate in a similar manner during the isothermal evaporation so that the shape of the factor thermogram remains more or less constant between conditions. However, the occurrence of aqueous-phase processes may complicate the grouping of compounds especially for highly oxidized samples (Buchholz et al., 2020). Compounds with somewhat different volatility may no longer be separated but rather be grouped together due to how they are affected by the aqueous phase. This can create changes in the appearance of the factor thermogram (e.g., broadening) and possibly induce a non-negligible shift in $T_{desorp}$ ($\geq 15$ °C) dependent on the extent of aqueous-phase processes. We provide more details about the behavior of the PMF algorithm, how compounds are grouped and why the shape and characteristic $T_{desorp}$ may change in the Supplement (see Section S1.2.3).

Comment       L215: It would be useful to have further discussion of how the "background" factors were identified. The authors retain only 5 factors in their analysis, suggesting that there are 5 background factors. This seems excessive. What does it mean for these to be "predominately in filter blank measurements?"

Response      We agree with the raised comment. As shown in Figure R4, we compared the total sum of sample and background factors in each FIGAERO-CIMS sample for α-pinene (a) and SQTmix SOA particles (b), respectively. In each SOA system, the absolute signal strength of the total sum of background factors is similar regardless of sample types. Within filter blank samples, background factors account for at least 80% of the total sum signal. Moreover, background factors displayed either constant or very shallow thermal desorption profiles. We modified the paragraph about the background factors to clarify the identification of factor types in section 3.2.

We would also like to point out that we did not identify 5 background factors in each sample. Instead, there are two sets of background factors. This may be due to changes in contamination when changing/inspecting the FIGAERO filter between experiments. Each blank measurement has 2 or 3 B factors. For the SQTmix SOA, B3 occurred in each blank. But Snap Blank 54 contained B2, while the Snap Blanks 60 and 61 had a somewhat different factor B1. The sum of B3 & B2, and B3 & B1 were almost identical. The samples collected on the corresponding days also contained either B1 or B2. This highlights the importance of regular blank measurements as close to the experiment conditions as possible. Also, this shows how the PMF analysis still identifies the background contributions even if there are some changes in the detailed composition.

While these details are very interesting and taught us a lot about the FIGAERO sample collection and how to improve the sample quality especially for very small mass loadings (< 30ng), this is besides the main focus of this paper. Hence, we decided to not include a detailed interpretation of the background factors in the manuscript.

[Figure]

**Figure R4**. Total sum of sample factors (i.e., type F) and background factors (i.e., type B) in FIGAERO-CIMS samples for α-pinene (a) and SQTmix SOA particles (b).

Change          Section 3.2

[…] The sum of type B factors showed similar absolute signal strength regardless of sample types. But while this contributed 10 - 60% to the total sum signal of the particle samples, it accounted for more than 80% of the total sum signal in filter blank samples. Type B factors displayed either nearly constant or very shallow factor thermograms.

**Comment**    Fig. S6: I find these "Kroll diagrams" a bit strange. These assume, presumably, that every ion is a unique molecule and not a fragment, correct? It could be more informative (or at least equally informative) to show one diagram that uses the average OsC and average Cnum for each factor.

**Response**    We do not assume that every ion is a unique molecule and not a fragment. Instead, every ion can be multiple molecules or a fragment due to thermal decomposition. Using these "Kroll diagrams" in Figure S6 gave us useful information beyond the average properties of factors and allowed us to compare ion distributions between sample factors. Different from the traditional Kroll diagram, the modified one used in Figure S6 (now renumbered as Figure S10 in the supplement) avoided the overlapping issue which arises from hundreds of molecules detected on CIMS but still gave a better visualization by lumping compounds with the same carbon number into grids with a 0.2-interval on the y-axis of OSc. We have modified the introduction of these diagrams at the end of Section 3.2 to avoid further misunderstandings.

We agree that it can be helpful to show one diagram summarizing the OSc and Cnum info for each factor which is only presented in the factor in the labels of Figure 3c and 4c in the main text. The corresponding results are presented in Figure S11.

**Changes**    In section 3.2

[…] Furthermore, ion distributions and bulk properties are visualized for each sample factor in the form of modified Kroll diagrams (Kroll et al., 2011) in Figure S10 and S11 by plotting the average carbon oxidation state (OSc) versus the carbon number (Cnum). By lumping ions with the same carbon number into a grid with a 0.2-interval on the y-axis of OSc, the issue of overlapping signals was avoided.

[Figure]

**Figure S11**. Kroll diagrams for the average carbon oxidation state (OSc) and carbon number (Cnum) of each sample factor in α-pinene (circle) and SQTmix (square) SOA particles.

Comment     Definition of factors as V or D: I find it helpful that the authors have worked to classify factors according to their thermal profiles. I think that this helps with understanding. But it would be useful to hear more about how they made decisions in what might be considered in between or marginal cases. For example, for the AV4 α-pinene factor, the desorption profiles are really, really broad and with much intensity remaining out to very high temperatures. And the fresh high RH SV4 SQTmix factor looks pretty similar to the SD5 factor for high RH RTC. As such, there seems to be some ambiguity in the definitions/assignment and it would be helpful if the authors were to address this further.

Response     In light of the insights we gained from composing the answer to the comments about the shape and characteristic T$_{desorp}$ changes of the factor thermograms, we decided to abandon the sharp distinction between type V and D in the manuscript and only refer to "sample" factors (type F) vs background factors (type B). This is only a change in terminology to make the discussion more readable. We do keep the detailed description about the behavior of AF5 and SF5 (before AD5 and SD5) and how that is different from the factors AF1-4 and SF1-4 in sections 3.2.1 and 3.2.2. However, since the reviewer raises a legitimate question about the differences between SF4 and SF5, we provide the following answer anyway. In the reply we use the factor labels of the revised manuscript (AV4 = AF4, AD5 = AF5, SV4 = SF4, SD5 = SF5).

Whether a factor is considered to be type V or D is based on the careful analysis of its thermal desorption behavior and mass spectrum. Since no absolutely objective parameters can be used, there are borderline cases which could fall into either category. The classification with D or V is mostly there to indicate the dominance of thermal decomposition and the disconnect of molecular weight and elemental composition trends with the factor volatility (see details below for SF4 & SF5). Both types were included in the analysis and interpretation of the PMF results.

The desorption profile of AF4 was very broad and remained at high intensity at very high temperature. But it still contained characteristic features which are distinguishable from the decomposition factor AF5. AF4 displayed a distinguishable maximum desorption temperature in its desorption profile. Additionally, we grouped compounds by carbon number and calculated their signal contribution to each sample factor, shown in Figure R5. As compared to any other factor, AF4 had the largest contribution of compounds with carbon number of 10 or more and thus had the highest average molecular weight. Note that AF4 also consisted of ions with carbon number of 6 or less (approximately 20% in signal contribution) as well, which might indicate non-negligible contributions of decomposition products.

[Figure]

**Figure R5**. Signal distribution across three ranges of carbon number (Cnum) in sample factors of α-pinene SOA particles.

In the SQTmix SOA system, SF4 and SF5 did show very similar thermograms. However, the mass spectrum of SF4 displayed very different features from that of SF5. The mass spectrum of SF4 had a larger contribution of compounds above 200 amu. We again grouped compounds by carbon number and calculated their signal contribution to each sample factor, shown in Figure R5. For SF1 – 4, the contribution of compounds with carbon number of 11 or more increased with the decreasing factor volatility as indicated, for instance, by the characteristic $T_{desorp}$ (the $25^{th} – 50^{th} – 75^{th}$

percentile values) of a factor thermogram. However, this trend breaks, when applied to SF5. Even though SF5 had the $T_{desorp}$ as high as SF4, its factor mass spectrum rather resembles that of SF3. In other words, using the factor mass spectrum of SF5 to estimate the volatility with an elemental composition-based parameterization would yield a volatility similar to that of SF3 which is inconsistent with the differences in their $T_{desorp}$ values. While it is possible that the compounds with low carbon number in SF5 are isomers with much lower volatility, the more likely explanation is that these are fragments of much larger/heavier molecules which decomposed before they could evaporate.

[Figure]

**Figure R6**. Signal distribution across four ranges of carbon number (Cnum) in sample factors of SQTmix SOA particles.

Change        Section 3.2

Two types of factors were identified. Factors occurring in particle samples but predominantly in filter blank measurements are defined as type B ("background") factors. The sum of type B factors showed similar absolute signal strength regardless of sample types. But while this contributed 10 - 60% to the total sum signal of the particle samples, it accounted for more than 80% of the total sum signal in filter blank samples. Type B factors displayed either nearly constant or very shallow factor thermograms. Factors which showed contributions in particle samples but not in filter blank samples were assumed to describe the collected particle sample and thus defined as type F ("sample") factors. In Buchholz et al. (2020), these sample factors were distinguished into the ones dominated by direct desorption of compounds (type V) and those dominated by products of thermal decomposition (type D). The careful analysis of the sample factors in this study showed that we could not make such a strict distinction. Thus, we decided to use the terms background factor (type B) and sample

factor (type F) and point out which of the sample factors shows strong signs of thermal decomposition products.

**Comment**    L263: The authors note that it is difficult to compare between experiments in terms of absolute signals owing to differences and uncertainty in collected mass. However, the authors presumably know the volume of material collected, and using reasonable estimates of density this should allow for comparability to well within a factor of two. Through normalization to the total signal, the authors do work to make the factor-specific observations quantitatively comparable. Use of a second method to estimate the total mass collected would help to validate the normalization method, as there is an implicit (but as best I can tell unstated) assumption that the CIMS response is the same for all factors.

**Response**    The information about the used sensitivity values for the CIMS data analysis are indeed important. We have added the information about using uniform sensitivity and transmission for all ions now to the detailed description of the CIMS measurements in Supplement section S1.1.4:

**Change**    Section S1.1.4

In the absence of a reliable transmission and sensitivity calibration for the relevant compounds detected with FIGAERO-CIMS, we assume uniform sensitivity and transmission for all ions. We conducted SMPS measurements prior to the FIGAERO-CIMS sample collection. Accounting for the amount of particulate water, changes in the VFR and organic density, we derived the theoretical collected mass on the filter from the sampled volume for the fresh stages. For the RTC stages, particles in the 100-L RTC were assumed to be 100% collected on the FIGAERO filter to provide an upper limit of the collected amount. As shown in Figure R7, the collected mass is almost linearly correlated with the total signal of sample factor between conditions, which validates the normalization methods used in this study. The collected mass seems to be overestimated for the two high-RH RTC samples. These experiments proved to be the most challenging to collect sufficient material on the filter. However, we could not rule out if there were increased particle losses in the transfer from the RTC to the filter or if other issues led to the overestimation from the SMPS measurements.

[Figure]

**Figure R7**. Association between total sample signals on FIGAERO-CIMS and collected mass derived from SMPS measurements. The total sample signal is calculated as the product of signal intensity (I) and molecular weight (MW). This allows for a convenient cross-instrument comparison between FIGAERO-CIMS and SMPS. Unity sensitivity was applied for FIGAERO-CIMS data, while 100% filter collection efficiency was applied for deriving the collected mass from SMPS data. The error bar in the x axis indicates the range of collected mass. (Figure S2 in the supplement)

Comment   Fig. 5 and 6: I understand the authors' motivation to exclude $T_{desorp}$ and NCR values in cases for which the desorption profile is not well-formed owing to limited signal. However, for the NCR I would encourage the authors to put perhaps an open symbol at the lowest NCR value (0.1) as a visual indication to the reader that the signal is very small for that particular factor & condition.

Response   Thanks for the comment! We modified the figures as requested.

Comment   L300: Regarding the authors' discussion of SV4 as an ELVOC and the observation of a major shift in the NCR under humid conditions, I return to my previous comment regarding the identification of this factor as a "V" type to begin with. There is certainly, in my opinion, ambiguity in this identification, which affects the interpretation and determination that this observation is "surprising."

Further discussion here would be helpful. For SV2, I'm a little surprised myself to see this identified as being in the "ELVOC range." The peak of this factor profile is fully in the "LVOC" range. Further, the volatility really exists as a continuum with the sharp lines drawn for convenience more than reality (although there is, of course, a tie to

physical behavior that underlies these distinctions). Only once things go to high RH does the SV2 factor shift to the ELVOC volatility range. But this is a shift from the dry conditions, indicating that perhaps some chemical change has occurred.

Response     Regarding the assignment of type "V" to the SV4 (now SF4), please refer to our response above about the process for identifying type V vs D. The factor type identification does not affect our interpretation and the determination that this observation is "surprising". We use "surprising" to emphasize the importance of aqueous-phase processes for SF4 here. If volatility would be the only driving factor for the removal of SF4, little evaporation would be expected for SV4 within the timescale of 0.25 h (Li and Shiraiwa, 2019; Li et al., 2019).

It is indeed somewhat challenging to assign a factor to the certain (single) volatility range. Which volatility was assigned to the factor is carefully and objectively decided by the characteristic $T_{desorp}$ (the $25^{th}$ – $50^{th}$ – $75^{th}$ percentile values). We did not mean to state SF2 have volatility in the ELVOC range. The misunderstanding was due to the unclear use of the expression "(E)LVOC" without proper introduction. (E)LVOC indicates that factors are either in the LVOC or in the ELVOC range. We modified the sentence and also clearly specified the meaning of (E)LVOC at its very first occurrence.

Furthermore, as explained in the previous comment, we removed the strict distinction of type V and D and now just point out if a sample factor is dominated by products from thermal decomposition products.

Change       Section 3.1.2

Under dry conditions, a larger fraction of LVOC and ELVOC (collectively (E)LVOC) […]

Section 3.3.1

[…] The decrease of  NCR for SF2 and SF4, which have its volatility in the LVOC and ELVOC range respectively, […]

Comment      L330: The authors note that chemical transformations could cause changes in $T_{desorp}$ and thermogram widths with RH for a given factor. But what is a factor even if the chemical composition has changed? Isn't a factor presumably a collection of molecules (or at least ion signals) that are invariant/grouped together? If chemical

changes occur, wouldn't one expect this to end up as a different factor? Or is the argument here that chemical changes occur, but these chemical changes somehow lead to the same mass spectra/factor as before the chemical changes occurred? I suggest that further discussion is warranted. Similarly, on line 337 the authors note that "some of the compounds grouped into AV3 must have evaporated…or continued to react." Can it really be "some" of the compounds if the factor itself is a stable/real thing? (Yes, factors are just mathematical constructs, but they still need to be stable, right?) How does one lose some compounds but not others from a factor and still have it come out as the same factor?

Response    Please see our response to the comment above for Line 208.

Comment    Section 3.3.3: Here, the authors aim to provide general understanding of the nature of the chemical processes or shifts in volatility that occur upon humidification. One thing that I think could be useful is if the authors were to further compare between the α-pinene and SQTmix. For example, it seems noteworthy that the "D" type factor identified for both exhibit completely distinct behavior. If anything, I would have intuitively thought that the "decomposition" factors would exhibit similar behavior between both chemical systems. Clearly, intuition is not serving me well here, which is why I think that further compare/contrast would help to strengthen the discussion further.

Response    Thanks for the comment pointing out the need for a clarification. We would like to highlight that the molecular structures of the two precursors were very different which yielded quite distinctly different mass spectra for the two SOA systems. In total, 771 and 821 ions were identified in α-pinene and SQTmix SOA system. Only 338 common ions occurred in both SOA systems. This represents 53.6% and 43.8% of the total signals for α-pinene and SQTmix dry fresh samples which exhibited the least amount of isothermal evaporation.

Although both AF5 and SF5 showed up as decomposition factors, it does not indicate that their compositions are similar. As these two factors originated from two SOA system, it is highly possible that they can behave differently against particulate water. It is also important to remember in this context that the products of any decomposition process may be similar or even identical, but they may stem from completely different parent compounds. Especially, very small fragments (e.g., oxalic acid or acetic acid)

carry very little information about the original molecule they came from. We have now clarified this in the text.

Change        Section 3.3.2

[…] Although both AF5 and SF5 were dominated by products of thermal decomposition, it does not indicate that their compositions are similar. While the mass spectra of AF5 was dominated by ions with Cnum from 7 to 10, major ions in the mass spectra of SF5 tended to have Cnum of 6 or below (Figure S10). As these two factors originated from two different SOA systems, it is highly possible that they can behave differently against particulate water. It is also important to remember in this context that the products of any decomposition process may be similar or even identical, but they may stem from completely different parent compounds. Especially, very small fragments (e.g., oxalic acid or acetic acid) carry very little information about the original molecule they came from.

References

Buchholz, A., Lambe, A. T., Ylisirniö, A., Li, Z., Tikkanen, O.-P., Faiola, C., Kari, E., Hao, L., Luoma, O., Huang, W., Mohr, C., Worsnop, D. R., Nizkorodov, S. A., Yli-Juuti, T., Schobesberger, S., and Virtanen, A.: Insights into the o: C-dependent mechanisms controlling the evaporation of α-pinene secondary organic aerosol particles, Atmos. Chem. Phys., 19, 4061-4073, 2019.

Buchholz, A., Ylisirniö, A., Huang, W., Mohr, C., Canagaratna, M., Worsnop, D. R., Schobesberger, S., and Virtanen, A.: Deconvolution of FIGAERO–CIMS thermal desorption profiles using positive matrix factorisation to identify chemical and physical processes during particle evaporation, Atmos. Chem. Phys., 20, 7693-7716, 2020.

D'Ambro, E. L., Schobesberger, S., Zaveri, R. A., Shilling, J. E., Lee, B. H., Lopez-Hilfiker, F. D., Mohr, C., and Thornton, J. A.: Isothermal evaporation of α-pinene ozonolysis SOA: Volatility, phase state, and oligomeric composition, ACS Earth and Space Chemistry, 2, 1058-1067, 2018.

Kroll, J. H., Donahue, N. M., Jimenez, J. L., Kessler, S. H., Canagaratna, M. R., Wilson, K. R., Altieri, K. E., Mazzoleni, L. R., Wozniak, A. S., Bluhm, H., Mysak, E. R., Smith, J. D., Kolb, C. E., and Worsnop, D. R.: Carbon oxidation state as a metric for describing the chemistry of atmospheric organic aerosol, Nat. Chem, 3, 133-139, 2011.

Li, Y. and Shiraiwa, M.: Timescales of secondary organic aerosols to reach equilibrium at various temperatures and relative humidities, Atmos. Chem. Phys., 19, 5959-5971, 2019.

Li, Z., Tikkanen, O.-P., Buchholz, A., Hao, L., Kari, E., Yli-Juuti, T., and Virtanen, A.: Effect of decreased temperature on the evaporation of α-pinene secondary organic aerosol particles, ACS Earth and Space Chemistry, 3, 2775-2785, 2019.

Petters, S. S., Hilditch, T. G., Tomaz, S., Miles, R. E. H., Reid, J. P., and Turpin, B. J.: Volatility change during droplet evaporation of pyruvic acid, ACS Earth and Space Chemistry, 4, 741-749, 2020.

Pospisilova, V., Bell, D. M., Lamkaddam, H., Bertrand, A., Wang, L., Bhattu, D., Zhou, X., Dommen, J., Prevot, A. S., and Baltensperger, U.: Photodegradation of α-pinene secondary organic aerosol dominated by moderately oxidized molecules, Environ. Sci. Technol., 55, 6936-6943, 2021.

Schobesberger, S., D'Ambro, E. L., Lopez-Hilfiker, F. D., Mohr, C., and Thornton, J. A.: A model framework to retrieve thermodynamic and kinetic properties of organic aerosol from composition-resolved thermal desorption measurements, Atmos. Chem. Phys., 18, 14757-14785, 2018.

Yli-Juuti, T., Pajunoja, A., Tikkanen, O. P., Buchholz, A., Faiola, C., Vaisanen, O., Hao, L., Kari, E., Perakyla, O., Garmash, O., Shiraiwa, M., Ehn, M., Lehtinen, K., and Virtanen, A.: Factors controlling the evaporation of secondary organic aerosol from alpha-pinene ozonolysis, Geophys. Res. Lett., 44, 2562-2570, 2017.

Zaveri, R. A., Shilling, J. E., Zelenyuk, A., Zawadowicz, M. A., Suski, K., China, S., Bell, D. M., Veghte, D., and Laskin, A.: Particle-phase diffusion modulates partitioning of semivolatile organic compounds to aged secondary organic aerosol, Environ. Sci. Technol., 54, 2595-2605, 2020.

---

## Author Comment (AC2)

We gratefully thank all reviewers for the careful reading and valuable comments. Below we provide our point-by-point responses to the reviewer's comments. In the following context, **raised comments/suggestions** are marked in **black**, **responses** are presented in **red**, and **changes to the manuscript/supplement information** are indicated in **blue**. The figures and tables in the following response are numbered consecutively in three replies to reviewers. Additionally, we corrected any minor typo that we recognized in the manuscript and supplement.

The legitimate questions about the real meaning of the PMF factors in the context of thermogram data led us to reconsider the strict distinction of type V and D factors. Instead, we now use the term "sample" factor and relabel the factors as AF1 - AF5 (before AV1-AV4 and AD5) and SF1 - SF5 (before SV1-SV4 and SD5) for α-pinene and SQTmix SOA systems. We use the new labels in our responses to be consistent with the revised manuscript. Note that the interpretation of the factors has not changed, only the labels were adjusted to remove some potential for misunderstandings.

**Reply to Reviewer 2**

Li et al. conducted laboratory experiments to investigate the change in volatility and composition during the evaporation of SOA formed from alpha-pinene and a mixture of sesquiterpenes. They conducted two types of experiments, isothermal evaporation and thermo-desorption using the FIGAERO CIMS. They ran the experiments under different RH conditions to probe possible diffusive limitations and water-induced changes in composition. This study is well within the scope of the journal. However, there are some important missing pieces of information that affect the final conclusions. My comments are the following:

Comment      What were the mass concentrations of these experiments? How variable were the concentrations in dry and high RH experiments? It is unclear if the change in volatility (and composition) was due to the difference in mass loading which to the first order, determines the volatility distribution in the particles.

Response      We would like to thank the reviewer for bringing up the important issue. We didn't address the well enough in the text. For all evaporation experiments of one SOA system, the aerosol mass concertation in the OFR was very similar. Assuming a particle density of 1.4 g cm$^{-3}$, the mass loadings of polydisperse α-pinene and SQTmix SOA from the OFR were estimated to be $399 \pm 16$ and $128 \pm 16$ μg m$^{-3}$, respectively (Table R3).

Table R3. Summary of OFR Mass Concertation for α-pinene and SQTmix particle evaporation experiments

| Evaporation RH | OFR Mass Confrontation ($\mu g \ m^{-3}$)[a] | |
| --- | --- | --- |
| | α-pinene | SQTmix |
| Dry (< 7% RH) | 389 ± 9 | 112 ± 6 |
| Intermediate (40% RH) | 379 ± 10 | 123 ± 4 |
| High RH (80% RH) | 430 ± 16 | 149 ± 3 |

Note: [a]The OFR was always maintained at 40% RH for all evaporation experiments of one SOA system

For the α-pinene case, the mass concentration of organic material after size selection was 4.47 and 5.31 $\mu g \ cm^{-3}$ under dry and high RH conditions, respectively. For the SQTmix case, the corresponding values were 0.97 and 1.39 $\mu g \ cm^{-3}$ under dry and high RH conditions. For each SOA system of interest, the differences in mass concentration between dry and high RH conditions would not be large enough to significantly shift the volatility distribution of compounds in the condense phase.

The differences between dry and high RH conditions were caused by the necessary adjustment in experimental details and not by changes in the SOA production in the OFR. We have added the information on mass concentration of SOA particles after size selection to the text.

Change Section 2.1

For all evaporation experiments of one SOA system, the aerosol mass concertation in the OFR was very similar. Assuming a particle density of 1.4 $g \ cm^{-3}$, the mass concertation of polydisperse α-pinene and SQTmix SOA from the OFR was estimated to be 399 ± 16 and 128 ± 16 $\mu g \ m^{-3}$, respectively. It has been found that compounds with C* of 0.1 $\mu g \ m^{-3}$ and below dominates the SOA composition in a previous study using the same type of SOA (Ylisirniö et al., 2020). Even though the aerosol mass concentration in the OFR in our study is higher than the typical ambient level by one order of magnitude, such difference would not affect the gas-particle partitioning behavior of compounds with C* $\leq 0.1 \ \mu g \ m^{-3}$.

Section 3.1.2

For each SOA system of interest, similar mass concentration of organic material after size selection was ensured for both dry and high RH conditions so that the volatility

distribution of compounds in the condensed phase were not significantly affected. For the α-pinene case, the mass concentration of organic material after size selection was 4.47 and 5.31 μg cm$^{-3}$ under dry and high RH conditions, respectively. For the SQTmix case, the corresponding values were 0.97 and 1.39 μg cm$^{-3}$ under dry and high RH conditions.

**Comment**

Here particles were generated using an OFR, and the average O/C value (from Table S1) is on the high end compared to the O/C from most of the alpha-pinene SOA generated in chamber experiments. In fact, many studies on SOA viscosity were done using chamber SOA. The authors should discuss the effects of the highly oxidized (and polar) nature of the particles on the evaporation behavior and how would this affect the comparison to other studies.

**Response**

The O/C level is indeed high for the investigated SOA systems in this study. Current studies on SOA viscosity not only explore different types of chamber-generated SOA particles (Renbaum-Wolff et al., 2013; Maclean et al., 2021) but also investigate wide ranges of atmospheric relevant compounds with O/C as high as this study, such as 3-methylbutane-1,2,3-tricarboxylic acid (3-MBTCA, O/C = 0.75), levoglucosan (O/C = 0.83), sucrose (O/C = 0.92) and citric acid (O/C = 1.17) (Lienhard et al., 2015). Regarding the impact of O/C level on the particle evaporation, Buchholz et al. (2019) have suggested that increasing O/C levels overall makes SOA particles more resilient to evaporation. Under dry conditions, this is partly due to an increase in viscosity. But the decrease in isothermal evaporation is mostly caused by the decrease in volatility with increasing O/C level. Detailed process modelling showed that already at 40% RH SOA particles behave liquid-like, and kinetic limitations linked to high viscosity do not play a major role (Buchholz et al., 2019, Tikkanen et al., 2020, Li et al., 2019).

The biggest discrepancy between OFR-generated SOA and ambient/chamber SOA may be the fraction of organic hydro-peroxides. They may be formed in much larger fractions than usual due to the high HO$_2$ concentrations in the OFR, which will favor the respective path for RO$_2$ radicals (Peng et al., 2019). A higher (hydro-)peroxide fraction may be linked to some of the observed aqueous phase processes as was suggested in Buchholz et al. (2019). However, hydro-peroxide have been detected in ambient samples (Tong et al., 2021) and thus their behavior is relevant to better understand the processes linked to particle volatility and aqueous phase processes.

We have added a new paragraph to section 4 in which we put our work in context of other volatility studies and discuss the atmospheric relevance of our findings

Change        Section 4

To our knowledge, this is the first study investigating the volatility of SOA particles from a mixture of farnesene and bisabolene which are acyclic and monocyclic sesquiterpenes of atmospheric relevance. For α-pinene, multiple studies of isothermal evaporation at room temperature exist (Vaden et al., 2011; Wilson et al., 2014; Yli-Juuti et al., 2017; D'Ambro et al., 2018; Buchholz et al., 2019; Li et al., 2019; Zaveri et al., 2020; Pospisilova et al., 2021). However, even for this single precursor system, the formation conditions determine the isothermal evaporation behavior of the formed SOA and thus must be carefully considered when comparing different studies. The detailed composition of particles determines their volatility, viscosity, and behavior towards particulate water. Generally, particles containing increasing amounts of higher oxidized compounds will exhibit lower volatility (Buchholz et al., 2019; Zaveri et al., 2020; Pospisilova et al., 2021), but may be more likely to be susceptible to aqueous-phase reactions (Buchholz et al., 2019). Unfortunately, not all previous studies provide an O/C, OSc value or similar proxy to estimate the degree of oxidation, which makes further comparisons difficult.

Comment       Did the presence of moist on the FIGAERO filter in high RH experiments affect the thermograms? Did the authors conduct any tests to make sure that for a single compound, or a mixture of known compounds, high RH did not change the shape of the thermograms?

Response      This is indeed a relevant point. Unfortunately, we did not conduct any test with individual compounds to validate the humidity independence of thermograms, and at the moment we are not able to perform the measurement due to the deployment situation of our CIMS. We will explore this as soon as our FIGAERO-CIMS is available.

However, to address this comment using the existing data, we carefully compared the single ion thermograms of multiple ions under dry and high RH conditions for this study, as shown in Figure R8. We do not observe systematic shifts in $T_{desorp}$ for all compounds between two RH conditions. From this we conclude that there could not be a general change in the shape of the thermograms simply due to the higher RH.

Note that the inlet and filter material (PTFE) is extremely hydrophobic. A drop of water, placed on the filter with a syringe, does not soak into the filter but rather remains on the surface until it has evaporated. This proved to be a challenge for calibration purposes but means that it is unlikely that the filter itself becomes "moist".

[Figure]

**Figure R8**. Ion thermograms of compositions consistent with $C_7H_{10}O_6$ (a), $C_7H_{10}O_7$ (b), $C_{16}H_{22}O_8$ (c) and $C_{13}H_{24}O_{12}$ (d) in dry fresh (red), high RH fresh (blue) and DMA filter blank (grey) samples

Comment    How was the relationship between $T_{desorp}$ and volatility determined (Figure 2a)? Did the authors do any calibration using known compounds?

Response    The relationship between $T_{desorp}$ and volatility was derived by calibrating the FIGAERO-CIMS against a series of polyethylene glycols (PEG, chain length 5-8 units) particles with 80 nm electrical mobility diameter. The detail of the calibration procedure and results can be found in Ylisirniö et al. (2021). We refer to this method in the line 114 in the original manuscript and now added the calibration parameters to the Supplement.

Comment    What was the vapor wall loss in the residence time chamber? How did vapor wall loss affect particle volatility and composition in the experiments?

Response  The model simulation of a previous study using the same setup has suggested that the vapor wall losses in the RTC were fast with a vapor wall loss coefficient greater than $10^{-2}$ s$^{-1}$ (Yli-Juuti et al., 2017).

We additionally characterized our RTC with 80-nm octaethylene glycol (PEG8) particles under dry conditions at the same experimental temperature (294 K). The corresponding evapogram is presented in the Figure R9. Applying the eq 1 in Salo et al. (2010) with the parameters used by Krieger et al. (2018), we yielded a saturation vapor pressure of $1.35 \times 10^{-7}$ Pa for PEG8, which is consistent with the reported value of $9.2 \times 10^{-8}$ Pa at 298 K in Krieger et al. (2018). Thus, we conclude that the vapor wall loss is fast enough to keep the gas phase concentrations of organics negligible low in the RTC and therefore the vapor wall loss rate would not impact particle volatility and composition in the experiments.

[Figure]

**Figure R9**. Evapogram of 80-nm octaethylene glycol (PEG8) particles under dry condition at 294 K.

Comment  Based on Figure 3b, it seems that the authors did not observe O3 and O4 species that are known to be major a-pinene oxidation products (e.g., pinic acid, pinoic acid). What is the reason for that?

Response  We see how this misunderstanding happened. The O content of the individual molecules is not visible in Figure 3b. The chemical formulas in Figure 3b stand for the average composition of the sample factors instead of individual molecules. We did indeed observe O3 and O4 species. As shown in the Figure R10, signals of compositions corresponding to the three mentioned α-pinene oxidation products (i.e.,

pinic acid, pionic acid and norpinic acid) are clearly higher than the background noise. Note that these only representing the contribution to the particle phase. Especially, pinonic acid is expected to reside predominantly in the gas phase at the chosen conditions (Lopez-Hilfiker et al., 2015).

[Figure]

**Figure R10**. Ion thermograms of compositions consistent with pinic acid (a), pinonic acid (b) and norpinic acid (c) in dry fresh (red) and DMA filter blank (grey) samples

**References**

Buchholz, A., Lambe, A. T., Ylisirniö, A., Li, Z., Tikkanen, O.-P., Faiola, C., Kari, E., Hao, L., Luoma, O., Huang, W., Mohr, C., Worsnop, D. R., Nizkorodov, S. A., Yli-Juuti, T., Schobesberger, S., and Virtanen, A.: Insights into the o: C-dependent mechanisms controlling the evaporation of α-pinene secondary organic aerosol particles, Atmos. Chem. Phys., 19, 4061-4073, 2019.

D'Ambro, E. L., Schobesberger, S., Zaveri, R. A., Shilling, J. E., Lee, B. H., Lopez-Hilfiker, F. D., Mohr, C., and Thornton, J. A.: Isothermal evaporation of α-pinene ozonolysis SOA: Volatility, phase state, and oligomeric composition, ACS Earth and Space Chemistry, 2, 1058-1067, 2018.

Krieger, U. K., Siegrist, F., Marcolli, C., Emanuelsson, E. U., Gøbel, F. M., Bilde, M., Marsh, A., Reid, J. P., Huisman, A. J., and Riipinen, I.: A reference data set for validating vapor pressure measurement techniques: Homologous series of polyethylene glycols, Atmos. Meas. Tech., 11, 49-63, 2018.

Li, Z., Tikkanen, O.-P., Buchholz, A., Hao, L., Kari, E., Yli-Juuti, T., and Virtanen, A.: Effect of decreased temperature on the evaporation of α-pinene secondary organic aerosol particles, ACS Earth and Space Chemistry, 3, 2775-2785, 2019.

Lienhard, D. M., Huisman, A. J., Krieger, U. K., Rudich, Y., Marcolli, C., Luo, B., Bones, D. L., Reid, J. P., Lambe, A. T., and Canagaratna, M. R.: Viscous organic aerosol particles in the upper troposphere: Diffusivity-controlled water uptake and ice nucleation?, Atmos. Chem. Phys., 15, 13599-13613, 2015.

Lopez-Hilfiker, F., Mohr, C., Ehn, M., Rubach, F., Kleist, E., Wildt, J., Mentel, T. F., Carrasquillo, A., Daumit, K., and Hunter, J.: Phase partitioning and volatility of secondary organic aerosol components formed from α-pinene ozonolysis and OH oxidation: The importance of accretion products and other low volatility compounds, Atmos. Chem. Phys., 15, 7765-7776, 2015.

Maclean, A. M., Smith, N. R., Li, Y., Huang, Y., Hettiyadura, A. P., Crescenzo, G. V., Shiraiwa, M., Laskin, A., Nizkorodov, S. A., and Bertram, A. K.: Humidity-dependent viscosity of secondary organic aerosol from ozonolysis of β-caryophyllene: Measurements, predictions, and implications, ACS Earth and Space Chemistry, 5, 305-318, 2021.

Peng, Z., Lee-Taylor, J., Orlando, J. J., Tyndall, G. S., and Jimenez, J. L.: Organic peroxy radical chemistry in oxidation flow reactors and environmental chambers and their atmospheric relevance, Atmos. Chem. Phys., 19, 813-834, 2019.

Pospisilova, V., Bell, D. M., Lamkaddam, H., Bertrand, A., Wang, L., Bhattu, D., Zhou, X., Dommen, J., Prevot, A. S., and Baltensperger, U.: Photodegradation of α-pinene secondary organic aerosol dominated by moderately oxidized molecules, Environ. Sci. Technol., 55, 6936-6943, 2021.

Renbaum-Wolff, L., Grayson, J. W., Bateman, A. P., Kuwata, M., Sellier, M., Murray, B. J., Shilling, J. E., Martin, S. T., and Bertram, A. K.: Viscosity of alpha-pinene secondary organic material and implications for particle growth and reactivity, Proc. Natl. Acad. Sci. U. S. A., 110, 8014-8019, 2013.

Salo, K., Jonsson, Å. M., Andersson, P. U., and Hallquist, M.: Aerosol volatility and enthalpy of sublimation of carboxylic acids, The Journal of Physical Chemistry A, 114, 4586-4594, 2010.

Tong, H., Liu, F., Filippi, A., Wilson, J., Arangio, A. M., Zhang, Y., Yue, S., Lelieveld, S., Shen, F., and Keskinen, H.-M. K.: Aqueous-phase reactive species formed by fine particulate matter from remote forests and polluted urban air, Atmos. Chem. Phys., 21, 10439-10455, 2021.

Vaden, T. D., Imre, D., Beranek, J., Shrivastava, M., and Zelenyuk, A.: Evaporation kinetics and phase of laboratory and ambient secondary organic aerosol, Proc. Natl. Acad. Sci. U. S. A., 108, 2190-2195, 2011.

Wilson, J., Imre, D., Beránek, J., Shrivastava, M., and Zelenyuk, A.: Evaporation kinetics of laboratory-generated secondary organic aerosols at elevated relative humidity, Environ. Sci. Technol., 49, 243-249, 2014.

Yli-Juuti, T., Pajunoja, A., Tikkanen, O. P., Buchholz, A., Faiola, C., Vaisanen, O., Hao, L., Kari, E., Perakyla, O., Garmash, O., Shiraiwa, M., Ehn, M., Lehtinen, K., and Virtanen, A.: Factors controlling the evaporation of secondary organic aerosol from alpha-pinene ozonolysis, Geophys. Res. Lett., 44, 2562-2570, 2017.

Ylisirniö, A., Barreira, L. M. F., Pullinen, I., Buchholz, A., Jayne, J., Krechmer, J. E., Worsnop, D. R., Virtanen, A., and Schobesberger, S.: On the calibration of FIGAERO-tof-CIMS: Importance and impact of calibrant delivery for the particle-phase calibration, Atmos. Meas. Tech., 14, 355-367, 2021.

Ylisirniö, A., Buchholz, A., Mohr, C., Li, Z., Barreira, L., Lambe, A., Faiola, C., Kari, E., Yli-Juuti, T., Nizkorodov, S. A., Worsnop, D. R., Virtanen, A., and Schobesberger, S.: Composition and volatility of secondary organic

aerosol (SOA) formed from oxidation of real tree emissions compared to simplified volatile organic compound (VOC) systems, Atmos. Chem. Phys., 20, 5629-5644, 2020.

Zaveri, R. A., Shilling, J. E., Zelenyuk, A., Zawadowicz, M. A., Suski, K., China, S., Bell, D. M., Veghte, D., and Laskin, A.: Particle-phase diffusion modulates partitioning of semivolatile organic compounds to aged secondary organic aerosol, Environ. Sci. Technol., 54, 2595-2605, 2020.

---

## Author Comment (AC3)

We gratefully thank all reviewers for the careful reading and valuable comments. Below we provide our point-by-point responses to the reviewer's comments. In the following context, **raised comments/suggestions** are marked in **black**, **responses** are presented in **red**, and **changes to the manuscript/supplement information** are indicated in **blue**. The figures and tables in the following response are numbered consecutively in three replies to reviewers. Additionally, we corrected any minor typo that we recognized in the manuscript and supplement.

The legitimate questions about the real meaning of the PMF factors in the context of thermogram data led us to reconsider the strict distinction of type V and D factors. Instead, we now use the term "sample" factor and relabel the factors as AF1 - AF5 (before AV1-AV4 and AD5) and SF1 - SF5 (before SV1-SV4 and SD5) for α-pinene and SQTmix SOA systems. We use the new labels in our responses to be consistent with the revised manuscript. Note that the interpretation of the factors has not changed, only the labels were adjusted to remove some potential for misunderstandings.

**Reply to Reviewer 3**

This work by Li et al. investigates the changes in volatility and composition of SOA from the oxidation of α-pinene and a sesquiterpene mixture as a result of isothermal evaporation. It details laboratory experiments performed with a FIGAERO-CIMS to obtain volatility information and molecular formulas for species in the SOA. Results from the sesquiterpene mixture were compared to that from α-pinene, and also compared between dry (RH <7%) and wet (RH ~80%) conditions. Positive matrix factorization (PMF) was used to investigate the behavior of individual factors before and after evaporation at the two RH extremes. Unsurprisingly, one result is that the sesquiterpene SOA is more resistant to evaporation than α-pinene, implying lower volatility. They also found that under high RH, more of the signal evaporated. This work is novel, will be of interest to the readers of ACP, and provides valuable information to the community. After addressing the minor general and specific comments below, this manuscript will be suitable for publication in ACP.

**General comments**

Comment    The use of so many acronyms make the manuscript difficult to follow at times. Some examples include:

- the abbreviation of α-pinene to αpin is unnecessary

We have removed the abbreviation "apin" and replaced that with "α-pinene" through the text.

- changing STG to ΣTG could make it easier for the reader to remember what this stands for

  We do not agree with changing STG to ΣTG for our manuscript. Since we have used subscripts to indicate normalization or scaling for different versions of STG in the section 2.2, using ΣTG would contradict with the original ideas of normalized and scaled STG. Furthermore, we are convinced that mixing capital Greek and Roman letters in this case will instead decrease the readability.

- RTC for residence time chambers is used to refer to both the physical chamber (lines 94-96, 105, 111) and the experiment type (lines 130, 160, 182, etc) which is confusing. When referring to experiment type does it always refer to a specific evaporation time length?

  We disagree with the assumption that the dual use of RTC causes major problems in this context. The label "RTC" for the sample type indicates that the particles were inside the RTC. The "fresh" particles never entered the RTC.

  The labels "fresh" and "RTC" for the two evaporation stages always refer to the same evaporation times as stated in the manuscript (e.g., line 182 original version)

- CNerror is only used once (line 155).

  We have removed the abbreviation CNerror.

**Comment** This study could benefit from a more robust discussion of, and comparison to, the several previous studies measuring the evaporation of α-pinene SOA

**Response** We agree that it is important to put new findings in the context of existing studies. Unfortunately, it was difficult to find truly comparable studies for our experiments. We could not find any volatility measurements for SOA from acyclic SQTs that were similar to our SQTmix. For α-pinene, multiple studies of isothermal evaporation at room temperature exist (Vaden et al., 2011; Wilson et al., 2014; Yli-Juuti et al., 2017; D'Ambro et al., 2018; Buchholz et al., 2019; Li et al., 2019; Zaveri et al., 2020; Pospisilova et al., 2021). However, even for a single precursor system, the SOA formation conditions determine the volatility of the formed SOA. Buchholz et al. (2019) show that the VFR( $t_R$=4h) of α-pinene SOA particles with a wide range of OSc (i.e., from -0.46 to 0.63) is between 0.45 and 0.90. Pospisilova et al. (2021) also show that the degree of photochemical ageing changes the observed isothermal evaporation under dry conditions. In their study, the OSc was between -0.8 and -0.4 (estimated

from $f_{43}$ and $f_{44}$ values) and the VFR values at 0.4 to 0.8. Li et al. (2019) showed that the observed isothermal evaporation and the sensitivity to particulate water is different for α-pinene SOA of similar OSc formed from either photooxidation or ozonolysis. D'Ambro et al. (2018) investigated isothermal evaporation of particles collected on a FIGAERO filter. They showed that SOA with different physical age (i.e., produced under similar conditions in different simulation chambers) can exhibit different volatility and apparent composition.

Although there are multiple studies of isothermal evaporation of α-pinene SOA particles, very few studies provide molecular information that is comparable to our approach. D'Ambro et al. (2018) conducted FIGAERO-CIMS measurements of particles that were allowed to evaporate on the filter after collection. Although the α-pinene SOA particles in the study may not be directly comparable to the particles in our study, some of their findings share similarities with the interpretation of our PMF factors. For each ion, they explain the observed isothermal evaporation behavior with a model containing 3 components with different apparent volatility: (i) free monomers that evaporate from particles according to their C* values, (ii) ELVOC compounds that do not evaporate from the particles at room temperature but decompose upon heating to be detected as the single ion, and (iii) reversible oligomers that decompose into the corresponding free monomers with time or heat. In our data set, many individual ions show contributions from multiple factors. AF1 and SF1 are predominantly containing compounds that behave like "free monomers". AF5 and SF5 are mostly ELVOC compounds that are detected as thermal decomposition products. The behavior described for "reversible oligomers" is in line with the complex behavior of the PMF factors which we associate with aqueous-phase processes. As D'Ambro et al. (2018) only applied their model investigation to particle evaporation at 50% RH and above, it is impossible to determine whether the particle phase processes affecting the reversible oligomers are linked to the presence of particulate water. Note that the approach of D'Ambro et al. (2018) deploys a ion-by-ion model fitting, while our PMF analysis inspects the behavior of all ions in the data set at once.

We added the summary of this to the appropriate place in section 3.3.3 and 4 of the manuscript to give a stronger context for our findings.

Change          Section 3.3.3

Although there are multiple studies of isothermal evaporation of α-pinene SOA particles, very few studies provide molecular information that is comparable to our approach. D'Ambro et al. (2018) conducted FIGAERO-CIMS measurements of particles that evaporated on the filter after collection. Although the α-pinene SOA particles in the study may not be directly comparable to the particles in our study, some of their findings share similarities with the interpretation of our PMF factors. For each ion, they explain the observed isothermal evaporation behavior with a model containing 3 components with different apparent volatility: (i) free monomers that evaporate from particles according to their C* values, (ii) ELVOC compounds that do not evaporate from the particles at room temperature but decompose upon heating to be detected as the single ion, and (iii) reversible oligomers that decompose into the corresponding free monomers with time or heat. In our data set, many individual ions show contributions from multiple factors. AF1 and SF1 are predominantly containing compounds that behave like "free monomers". AF5 and SF5 are mostly ELVOC compounds that are detected as thermal decomposition products. The behavior described for "reversible oligomers" is in line with the complex behavior of the PMF factors which we associate with aqueous-phase processes. As D'Ambro et al. (2018) only applied their model investigation to particle evaporation at 50% RH and above, it is impossible to determine whether the particle phase processes affecting the reversible oligomers are linked to the presence of particulate water. Note that the approach of D'Ambro et al. (2018) deploys a ion-by-ion model fitting, while our PMF analysis inspects the behavior of all ions in the data set at once.

Section 4

To our knowledge, this is the first study investigating the volatility of SOA particles from a mixture of farnesene and bisabolene which are acyclic and monocyclic sesquiterpenes of atmospheric relevance. For α-pinene, multiple studies of isothermal evaporation at room temperature exist (Vaden et al., 2011; Wilson et al., 2014; Yli-Juuti et al., 2017; D'Ambro et al., 2018; Buchholz et al., 2019; Li et al., 2019; Zaveri et al., 2020; Pospisilova et al., 2021). However, even for this single precursor system, the formation conditions determine the isothermal evaporation behavior of the formed SOA and thus must be carefully considered when comparing different studies. The detailed composition of particles determines their volatility, viscosity, and behavior

towards particulate water. Generally, particles containing increasing amounts of higher oxidized compounds will exhibit lower volatility (Buchholz et al., 2019; Zaveri et al., 2020; Pospisilova et al., 2021), but may be more likely to be susceptible to aqueous-phase reactions (Buchholz et al., 2019). Unfortunately, not all previous studies provide an O/C, OSc value or similar proxy to estimate the degree of oxidation, which makes further comparisons difficult.

**Specific comments**

| | |
|---|---|
| Comment | Line 78: I think it could be helpful to the reader to 1) specify the mixture of sesquiterpenes, at least with respect to the most dominant species present, and 2) show their structures somewhere. |
| Response | We agree with the reviewer and added a sentence to clarify the composition of the sesquiterpene mixture. |
| Change | Section 2.1 |
| | […] Farnesene isomers (40%, acyclic) and bisabolene isomers (40%, monocyclic) are the two dominant species in the mixture of sesquiterpene, followed by other unidentified sesquiterpenes (Ylisirniö et al., 2020). […] |
| Comment | Paragraph starting line 78: please add your SOA loadings to this paragraph. If there was much more than typical ambient (>~50 ug/m3), can you please comment in the results and discussion section how your results may or may not be affected from partitioning more SVOC to the SOA than would be observed in ambient? |
| Response | The polydisperse SOA mass loading is indeed higher than typical ambient aerosol concentration ($399 \pm 16$ and $128 \pm 16$ µg m$^{-3}$, for α-pinene and SQTmix respectively). However, our previous study investigated the same type of SOA and found that the majority of compounds were within the range of LVOC and ELVOC. Such volatility range would not be affected by the mass difference between this study and typical ambient concentration. We added this information and our reasoning of why this does not impact the relevance of our findings in section 2.1. |
| Change | Section 2.1 |
| | […] For all evaporation experiments of one SOA system, the aerosol mass concertation in the OFR was very similar. Assuming a particle density of 1.4 g cm$^{-3}$, the mass concertation of polydisperse α-pinene and SQTmix SOA from the OFR was |

estimated to be $399 \pm 16$ and $128 \pm 16$ µg m$^{-3}$, respectively. It has been found that compounds with C* of 0.1 µg m$^{-3}$ and below dominates the SOA composition in a previous study using the same type of SOA (Ylisirniö et al., 2020). Even though the aerosol mass concentration in the OFR in our study is higher than the typical ambient level by one order of magnitude, such difference would not affect the gas-particle partitioning behavior of compounds with C* $\leq$ 0.1 µg m$^{-3}$. […]

| Comment | Line 81: Do you think forming the SOA at one RH and evaporating it at a different RH will affect your results? Do you think there is a long or short equilibration time between formation and evaporation RH? |
|---|---|

Response   One of the main objectives of this study is to investigate the evaporation of SOA particles under different RH conditions. Generating the SOA at one RH guarantees the same initial composition for the follow-up isothermal evaporation under different RH conditions. Observed differences in evaporation are then linked to the impact of RH and not a potentially different starting composition.

The equilibration time between formation and evaporation RH is supposed to be very short. According to the Figure S8 in Bones et al. (2012), the half-time to dry 80-nm sucrose (note that sucrose has amorphous solid phase state at dry conditions at 293K) particles from initially 40% RH at 293 K is less than 1 s.

Comment   Line 93-96:

Was there anything in these evaporation chambers to absorb desorbing vapors (for example activated charcoal)? If not, do you think there's a role for re-partitioning?

Response   We did not use activated charcoal as a vapor absorbent. Instead, the polished stainless wall acted as a perfect sink for the vapors. The model simulation of a previous study using the same setup suggested that the vapor wall losses in the RTC were fast with a vapor wall loss coefficient greater than $10^{-2}$ s$^{-1}$ (Yli-Juuti et al., 2017).

It is unlikely that re-partitioning takes place in our setup. We additionally characterized our RTC with 80-nm octo-ethylene glycol (PEG8) particles under dry conditions at the same experimental temperature (294 K). The corresponding evapogram is presented in the Figure R11. Applying the eq 1 in Salo et al. (2010) with the parameters used by Krieger et al. (2018), we yielded a saturation vapor pressure of $1.35 \times 10^{-7}$ Pa for PEG8, which is consistent with the reported value of $9.2 \times 10^{-8}$ Pa

at 298 K in Krieger et al. (2018). Thus, we conclude that the vapor wall loss would not impact particle volatility and composition in the experiments.

[Figure]

**Figure R11**. Evapogram of 80-nm octaethylene glycol (PEG8) particles under dry condition at 294 K.

| Comment | The workflow isn't obvious—were all particles sent sequentially through each stage laid out in i-iii, or did particles only go into one evaporation chamber? If particles only went into one type of reaction chamber, was the reasoning behind having 3 types to achieve different lengths of evaporation time? |
|---|---|
| Response | Yes, we had 3 different paths for the SOA samples to achieve different lengths of evaporation time. All particles were sent to only one of the three stages listed as i – iii. We corrected the corresponding sentence by changing "and" to "or". |
| Comment | Do you expect the same evaporation behavior and results in each chamber (i.e. no impacts from air volume to wall surface area, etc)? |
| Response | We compared the VFR values measured at the longest residence times (i.e., 40 min) in the mini-RTC with the corresponding points in the large size RTC. They agreed within the experimental uncertainties. |
| | For the large RTC, we tested if the mass loading inside the chamber affects the observed evaporation. No changes were observed for particle mass concentrations between 0.5 and 2.5 ug m$^{-3}$ (50 – 250 ng organic material inside the RTC). This agrees with the model calculations which found fast uptake of vapors on the walls and no saturation effects for this mass loading range (Yli-Juuti et al., 2017). |

The typical mass loading were 50 ng and 12.5 ng in the large and mini-RTC in the present study.

Comment    Line 94: "*a* 25 L… chamber*s*"— was one chamber used or multiple? If multiple, were they all identical?

Response    It should be singular. We corrected the typo.

Comment    Line 115: saturation vapor *pressure* or *concentration*?

Response    We decided to use saturation vapor concentration in the revised manuscript for consistency.

Comment    Line 120: instead of "the appearance", "the shape" might be more easily understood, but ok as-is.

Response    We still go with the appearance.

Comment    Line 145: half the sum thermogram signal or mass?

Response    It should be signal, and we clarify it by adding 'signal'.

Change    Section 2.2

[…] ($T_{50}$, at which half of the cumulative STG(T) signal is reached) […].

Comment    Line ~190: Can you please discuss why the overall volatility is lower under wet versus dry conditions, yet under these low-volatility wet conditions more signal is lost after evaporation?

Response    The particles collected on the FIGAERO filter are the residual particles after evaporation. If more of the more volatile compounds are removed before collection (i.e., VFR is lower), the remaining material is of lower volatility. Also, the amount of evaporated material does not necessarily predict the volatility of the remaining particles. One can imagine a sample with 20% SVOC and 80% ELVOC material vs a second sample with 50% of each. If the volatility distribution of the ELVOC fraction is the same, the STG after removal of the SVOC fraction will be the same for both samples. But the VFR will be very different (0.8 vs 0.5).

Comment    Line 200: Couldn't this be tested by comparing the signal to mass concentration under dry versus wet conditions? If the mass concentration is equivalent, you could determine what percent of signal was lost presumably due to evaporation of high volatility material during the SOA collection time. Or if the mass concentration was

different but the signal was the same, you'd have a hint that the SOA under the two conditions may have been made via two different pathways (e.g. reactive uptake at high RH vs. vapor-pressure driven condensation at low RH).

Response    Neither reactive uptake nor vapor pressure driven condensation should play a major role during sampling the monodisperse aerosol onto the filter. Note that gas-phase compounds were diluted by orders of magnitudes during the size selection step. Instead of vapor uptake, the evaporation during the sample collection will take place, as discussed in section 2.1. The small amounts of compounds evaporating from the particles are quickly removed by the stainless-steel tubing walls or will be removed with the sample flow.

Comment    Line 239: I think it makes sense to reference some of the papers on thermal decomposition upon heating with a FIGAERO here (or above).

Response    We add a reference (D'Ambro et al., 2018; Schobesberger et al., 2018; Yang et al., 2021) as requested.

Comment    Line 255: Would help to directly reference the figure in place of saying "above"

Response    Now we refer to the corresponding figures instead of using "above".

Comment    Line 265: excl = excluding? Please write out complete word for clarity

Response    We exchanged excl with excluding throughout the manuscript.

Comment    Line 270: rational should be rationale

Response    We changed it as requested.

Comment    Line 285: "decreases with *evolving* isothermal…" or "with *increasing* isothermal…"?

Response    We changed it as requested.

Comment    Line 287: if the NCR doesn't decrease with decreasing VFR couldn't this also mean that nothing is happening to the compounds, that they are neither being lost or produced?

Response    Yes, if NCR does not change, the compounds in this factor either do not change or the loss and production are balance. However, in this specific sentence, we were trying to describe the implication of "complex" behavior of NCR. With complex we mean that instead of a simple decrease, the NCR values show a "zigzag" patter for samples with

increasing isothermal evaporation. We have adjusted this sentence to avoid this misunderstanding.

Change          Section 3.3

On the other hand, complex behavior of the NCR with increasing isothermal evaporation (e.g., a decrease followed by an increase) indicates that the main loss mechanism of the compounds is likely chemical transformation.

Comment         line 303: can you speculate on other possible loss mechanisms?

Response        The other possible loss mechanism is aqueous-phase reaction. We discussed this in section 3.3.3. We clarified the existing section by adding "(i.e., aqueous-phase process)".

Change          Section 3.3.1

[…] (i.e., aqueous-phase process) […]

Comment         Line 307: Is it possible that there is different gas-phase chemistry or SOA formation pathways (i.e. reactive uptake vs. condensation) at high RH, so these species don't necessarily have to be produced in the particle phase?

Response        The RH in the OFR was always 40% to ensure that the initial composition of each SOA system was the same in the dry and high RH experiments. The change in RH occurred in the size selection step in the NanoDMA where also the gas phase is diluted. After this dilution, the concentrations in the gas phase are so low that no significant contribution back to the particle phase occurs. Instead, the particles evaporate. The evaporated compounds are quickly lost to the stainless-steel walls of the tubing and the RTC.

Comment         Line 319: figure 2, panels a & b only, correct?

Response        Yes. Now we add "a" and "b" for clarification

Comment         Line 331: figure 3, panel a only, correct? And figure 6 panel a?

Response        Yes. Now we add "a" for each figure.

Comment         Line 333: is it possible that there are more compounds grouped into the factor under wet conditions, instead of more signal of the same number of compounds?

Response        No. For each factor, the PMF-resolved factor mass spectrum does not change between conditions. This is the working principle of PMF. However, the residual (i.e.,

overestimation vs underestimation) may be different for compounds in the factor. For details, please see the answer to the second comment of reviewer #1 and the new Supplement section S1.2.3.

| | |
|---|---|
| Comment | Line 393: Should have a reference for α-pinene having largest emissions globally |
| Response | We add a reference (Guenther et al., 2012) as requested. |
| Comment | Line 397: sentence "These findings are generally…" is redundant |
| Response | We delete the sentence as requested. |
| Comment | Line 740/ Figure 2: Putting the y-axis in panels a & b on the same scale would make these easier to compare. |
| Response | Since we do not compare the same stage between different RH conditions, we decided to use the smaller scale for the panel (b) to enhance the smaller signal of the RTC stage. We do not compare the absolute signal intensity between the different RH conditions. Therefore, the same y-axis scale will not add information to this figure. |
| Comment | Key figure: is time on the x-axis for both the left and right figure/schematic? |
| Response | We apologize for the misunderstanding. The axis of the left panel is particle size while the right one is fraction. We add axis labels for both figures. The top-to-bottom orders are identical for figures on the two sides. |
| Change | |

**Evaporation of Sesquiterpene-mixed SOA Particles**

**Size Change**

**Molecular-level Change**

[Figure]

Size

Fraction (%)

**References**

Bones, D. L., Reid, J. P., Lienhard, D. M., and Krieger, U. K.: Comparing the mechanism of water condensation and evaporation in glassy aerosol, Proc. Natl. Acad. Sci. U. S. A., 109, 11613-11618, 2012.

Buchholz, A., Lambe, A. T., Ylisirniö, A., Li, Z., Tikkanen, O.-P., Faiola, C., Kari, E., Hao, L., Luoma, O., Huang, W., Mohr, C., Worsnop, D. R., Nizkorodov, S. A., Yli-Juuti, T., Schobesberger, S., and Virtanen, A.: Insights into the o: C-dependent mechanisms controlling the evaporation of α-pinene secondary organic aerosol particles, Atmos. Chem. Phys., 19, 4061-4073, 2019.

D'Ambro, E. L., Schobesberger, S., Zaveri, R. A., Shilling, J. E., Lee, B. H., Lopez-Hilfiker, F. D., Mohr, C., and Thornton, J. A.: Isothermal evaporation of α-pinene ozonolysis SOA: Volatility, phase state, and oligomeric composition, ACS Earth and Space Chemistry, 2, 1058-1067, 2018.

Guenther, A., Jiang, X., Heald, C. L., Sakulyanontvittaya, T., Duhl, T., Emmons, L., and Wang, X.: The model of emissions of gases and aerosols from nature version 2.1 (megan2. 1): An extended and updated framework for modeling biogenic emissions, Geoscientific Model Development, 5, 1471-1492, 2012.

Krieger, U. K., Siegrist, F., Marcolli, C., Emanuelsson, E. U., Gøbel, F. M., Bilde, M., Marsh, A., Reid, J. P., Huisman, A. J., and Riipinen, I.: A reference data set for validating vapor pressure measurement techniques: Homologous series of polyethylene glycols, Atmos. Meas. Tech., 11, 49-63, 2018.

Li, Z., Tikkanen, O.-P., Buchholz, A., Hao, L., Kari, E., Yli-Juuti, T., and Virtanen, A.: Effect of decreased temperature on the evaporation of α-pinene secondary organic aerosol particles, ACS Earth and Space Chemistry, 3, 2775-2785, 2019.

Pospisilova, V., Bell, D. M., Lamkaddam, H., Bertrand, A., Wang, L., Bhattu, D., Zhou, X., Dommen, J., Prevot, A. S., and Baltensperger, U.: Photodegradation of α-pinene secondary organic aerosol dominated by moderately oxidized molecules, Environ. Sci. Technol., 55, 6936-6943, 2021.

Salo, K., Jonsson, Å. M., Andersson, P. U., and Hallquist, M.: Aerosol volatility and enthalpy of sublimation of carboxylic acids, The Journal of Physical Chemistry A, 114, 4586-4594, 2010.

Schobesberger, S., D'Ambro, E. L., Lopez-Hilfiker, F. D., Mohr, C., and Thornton, J. A.: A model framework to retrieve thermodynamic and kinetic properties of organic aerosol from composition-resolved thermal desorption measurements, Atmos. Chem. Phys., 18, 14757-14785, 2018.

Vaden, T. D., Imre, D., Beranek, J., Shrivastava, M., and Zelenyuk, A.: Evaporation kinetics and phase of laboratory and ambient secondary organic aerosol, Proc. Natl. Acad. Sci. U. S. A., 108, 2190-2195, 2011.

Wilson, J., Imre, D., Beránek, J., Shrivastava, M., and Zelenyuk, A.: Evaporation kinetics of laboratory-generated secondary organic aerosols at elevated relative humidity, Environ. Sci. Technol., 49, 243-249, 2014.

Yang, L. H., Takeuchi, M., Chen, Y., and Ng, N. L.: Characterization of thermal decomposition of oxygenated organic compounds in FIGAERO-CIMS, Aerosol Science and Technology, doi: 10.1080/02786826.2021.1945529, 2021. 1-22, 2021.

Yli-Juuti, T., Pajunoja, A., Tikkanen, O. P., Buchholz, A., Faiola, C., Vaisanen, O., Hao, L., Kari, E., Perakyla, O., Garmash, O., Shiraiwa, M., Ehn, M., Lehtinen, K., and Virtanen, A.: Factors controlling the evaporation of secondary organic aerosol from alpha-pinene ozonolysis, Geophys. Res. Lett., 44, 2562-2570, 2017.

Ylisirniö, A., Buchholz, A., Mohr, C., Li, Z., Barreira, L., Lambe, A., Faiola, C., Kari, E., Yli-Juuti, T., Nizkorodov, S. A., Worsnop, D. R., Virtanen, A., and Schobesberger, S.: Composition and volatility of secondary organic aerosol (SOA) formed from oxidation of real tree emissions compared to simplified volatile organic compound (VOC) systems, Atmos. Chem. Phys., 20, 5629-5644, 2020.

Zaveri, R. A., Shilling, J. E., Zelenyuk, A., Zawadowicz, M. A., Suski, K., China, S., Bell, D. M., Veghte, D., and Laskin, A.: Particle-phase diffusion modulates partitioning of semivolatile organic compounds to aged secondary organic aerosol, Environ. Sci. Technol., 54, 2595-2605, 2020.